# Dynamic Relational Priming Improves Transformer in Multivariate Time Series

**Hunjae Lee** [1]  **Corey Clark** [1 2]

## Abstract

Standard attention in transformers employ static token representations that remain unchanged across all pair-wise computations in each layer. This limits their representational alignment with the potentially diverse dynamics of each token-pair interaction. While they excel in domains with relatively homogeneous relationships, standard attention may be inadequate in capturing heterogeneous inter-channel dependencies of multivariate time series (MTS) data where different channel-pair interactions within a single system may be governed by entirely different physical laws or temporal dynamics. To better align the attention mechanism for such domain phenomena, we propose attention with dynamic relational priming (prime attention). Prime attention modulates token representations for each token-pair, optimizing each pair-wise interaction for that specific relationship. Results demonstrate that prime attention consistently outperforms standard attention across benchmarks, achieving up to 6.5% improvement in forecasting accuracy. In addition, prime attention achieves comparable performance using up to 40% less sequence length compared to standard attention, demonstrating its superior relational modeling capabilities and potential for data efficiency. Code is available at https://github.com/timlee0131/Prime-Attention.

## 1. Introduction

An important challenge in applying transformers to multivariate time series (MTS) stems from domain mismatch. While transformers excel in domains like natural language processing (Bahdanau et al., 2014; Vaswani et al., 2017; Achiam et al., 2023; Devlin et al., 2019), computer vision (Guo et al., 2022; Carion et al., 2020; Dosovitskiy et al.,

2020), and graph representation learning (Yun et al., 2019; Joshi, 2025; Veličković et al., 2017; Rampášek et al., 2022) where relationships exhibit relatively homogeneous interaction patterns, MTS data can exhibit high degrees of heterogeneous inter-channel dependencies that resist such standardized processing. In language modeling, token relationships are predominantly semantic in nature, enabling most critical patterns to be captured by simple weighted sums of token representations. Similarly, in computer vision, spatial relationships dominate, enabling attention mechanisms to focus on regions of interest through uniform spatial reasoning. Learning on graphs exhibits comparable homogeneity, where node relationships are fundamentally structural and connectivity-based, allowing standard attention to model interactions through meaningful topological patterns (that are sometimes separated by relationship type (Schlichtkrull et al., 2018; Hu et al., 2020; Wang et al., 2019)). In these domains, representational uniformity aligns with the underlying relational homogeneity, enabling effective pattern discovery through weighted aggregation of static token representations. By static, we mean that token representations in each layer are fixed relative to all other tokens throughout pair-wise modeling. In other words, in standard attention, a given token presents the same representational perspective of itself across all of its pair-wise interactions with other tokens. We classify this property of standard attention mechanisms as *static relational learning*.

On the other hand, MTS data can exhibit qualitatively different heterogeneity within individual systems. Indeed, different token (or channel) pairs within the same system can display fundamentally different physical laws or temporal dynamics. These are distinctions that may transcend what can be captured through mere weighted sum of token representations in standard attention. For example, if token $A$ experiences lead-lag with token $B$ but has instantaneous correlations with token $C$, then the relational modeling process could benefit from token $A$'s representation being modulated differently for interacting with token $B$ than for token $C$ to ensure representational compatibility in each interaction. Standard attention cannot achieve this dynamic modulation of token $A$ and instead forces the same representational perspective of token $A$ on both tokens $B$ and $C$, potentially undermining the effective discovery of useful patterns that are unique to each pair-wise interaction.

[1]Department of Computer Science, Southern Methodist University, Dallas TX USA [2]SMU Guildhall. Correspondence to: Hunjae Lee <hunjael@smu.edu>.

*Proceedings of the 43$^{rd}$ International Conference on Machine Learning*, Seoul, South Korea. PMLR 306, 2026. Copyright 2026 by the author(s).

We propose attention with dynamic relational priming (prime attention) to better align the attention mechanism for the requirements of the MTS domain. Our dynamic priming modulates token representations for each specific pair-wise interaction, optimizing for computational alignment between tokens based on their relationship characteristics. We denote this property of prime attention as *dynamic relational learning* and formalize both static and dynamic relational learning in Section 3. When token $A$ interacts with token $B$, a learnable primer modulates $A$'s representation to align computationally with the $A$-$B$ relationship dynamics; when the same token $A$ interacts with token $C$, a different primer modulates $A$'s representation to better align with the distinct $A$-$C$ relationship requirements. This pair-specific priming process enables each interaction to access representational views specifically optimized for their unique relational patterns, transcending the one-size-fits-all constraints of standard attention and static relational learning. As such, prime attention's representational plasticity in pair-wise relational modeling allows for richer, more effective pattern discovery as tokens become better aligned to capture the unique dynamics within each pair.

Our proposed prime attention introduces marginal additional parameters (around 1.5% on top of standard attention) while providing transformers with the representational flexibility needed to capture rich, heterogeneous pair-wise dynamics that standard attention struggles with in MTS. Our empirical results in Section 6 show that prime attention consistently outperforms standard attention on state-of-the-art MTS transformers across benchmarks, achieving up to 6.5% improvement in forecasting accuracy. In particular, our experiments reveal that prime attention indeed shows largest improvements on heterogeneous datasets. We also show that prime attention, in certain instances, can use up to 40% less sequence length while achieving comparable (or slightly better) performance than standard attention, empirically demonstrating prime attention's superior relational modeling capabilities. In addition, we comapre prime attention against a selection of recent works in MTS that perform sophisticated relational learning beyond standard attention and find that prime attention consistently outperforms them as well.

## 2. Related Works

**Transformers in MTS**  Recent advances in MTS forecasting have demonstrated important progress toward relational modeling strategies. iTransformer (Liu et al., 2024) represents among the most performant transformers that explicitly model channel-wise relationships. Timer-XL (Liu et al., 2025b) similarly captures inter-channel dependencies using a transformer. FreDF (Wang et al., 2025) uses a standard channel-wise transformer as a backbone but also

considers model optimization in the frequency domain. Adjacent to channel-wise transformers, patch-based transformers (Zhang & Yan, 2023; Wang et al., 2024d; Lee & Clark, 2025) have also been successful in modeling inter-channel dependencies, tokenizing segments within each channel for more fine-grained interactions. While these methods have shown robust performance, their unifying limitation lies in their static relational modeling. There also exist works that avoid inter-channel interactions entirely, opting for a channel-independent (CI) approach. More information on CI models can be found in Section B.

**Other Relational Learning Methods in MTS**  Some recent works capture specific types of inter-channel relationships in more explicit ways. LIFT (Zhao & Shen, 2024) explicitly estimates lead-lag patterns between channels during pre-processing while LagTS (Liu et al., 2025a) represents a more unified approach, integrating lead-lag estimation into the learning process. LagTS first extracts relational information between pairs of channels through an MLP and employs a custom loss function to guide the learning of lagged correlations. However, this approach remains constrained by the static relational learning paradigm where channel representations remain invariant across all pair-wise interactions within each layer. Channel Clustering Module (CCM) (Chen et al., 2024) addresses relationship heterogeneity by employing learnable similarity metrics to group compatible channels, enabling cluster-specific processing that can capture shared patterns within channel groups. However, CCM's group-level processing potentially limits their ability to model unique dynamics between individual channel pairs that don't conform to cluster-wide patterns. Overall, while these approaches represent significant advances in relationship-aware modeling for MTS, they are architecturally constrained by their design focus on specific categories of inter-channel dependencies and static relational learning.

**Attention with Sharpened Coefficients**  Recent advances in attention mechanisms have increasingly focused on selective modulation of attention components to improve relational learning. However, current attention enhancement strategies are often fundamentally constrained by their sole focus on modulating attention coefficients. The differential transformer (Ye et al., 2025) subtracts the learned attention map with another, de-noising, attention map to systematically reduce noise accumulation and improve focus on relevant context. Selective attention (Leviathan et al., 2025) uses a cumulative mask on the attention map to achieve similar results. Similarly, rewiring for runway cascade (Lee & Clark, 2026) modulates attention coefficients to achieve more balanced information propagation in causal transformers. Likewise, a recent theoretical study on softmax function has formalized such enhancement strategies and proposes

a way to adaptively enhance attention coefficients as a potential solution (Veličković et al., 2025). In time series, SDformer (Zhou et al., 2024) applies a directional kernel to queries and keys to concentrate attention on the most informative variates. While enhancing attention coefficients by modulating the attention map has shown to improve performance and reduce noise accumulation particularly in the language domain, they still use static tokens to enhance attention. As such, they can broadly be seen as instances of standard attention with *sharpened* coefficients, certainly improving upon standard attention but still fundamentally constrained by the static relational learning paradigm as further explored in the next section. In a slight departure from these score-sharpening approaches, gated attention (Qiu et al., 2026) modulates the attention output with learnable, query-dependent gates rather than reshaping the attention map; yet because these gates scale a token's static representation rather than adapting it per interaction, gated attention likewise remains within the static relational learning paradigm.

## 3. Categorizing Relational Learning

Here, we formalize static and dynamic relational learning. We broadly refer to relational learning as methodologies that explicitly model interactions and dependencies between discrete entities in structured data, such as tokens in a set, nodes in a graph, or channels in MTS. In particular, we focus on dyadic, or pair-wise, relational learning.

Given a set of entities $\mathcal{X} = \{x_1, x_2, ..., x_N\}$, $x_i \in \mathbb{R}^d$, we define relational learning broadly as seeking to capture the interaction function $f(x_i, x_j) \rightarrow \mathbb{R}^d$ that characterizes the dependency between each pair $(x_i, x_j)$ for which $i$ and $j$ have some computational relationship. Our question of interest in relational learning is how the core representations of entities $(x_i, x_j) \ \forall i, j \in \{1, ..., N\}$ are conditioned during pairwise interactions in $f$.

**Definition 3.1** (Static relational learning). For any valid pair of entities $(x_i, x_j)$, the relational learning process $f$ is *static* if representations of $x_i$ and $x_j$ are invariant to each other.

Formally, for any pair $(x_i, x_j)$, the representations used in interaction are:

$$h_{i \rightarrow j} = \psi(x_i; \theta_i), \quad h_{j \rightarrow i} = \psi(x_j; \theta_j)$$

where $h_{i \rightarrow j}$ can be interpreted as representation of $x_i$ created for interaction with $x_j$ and vice versa. Then, the relational learning process is static if $\psi(\cdot; \theta_n)$ is independent of the interaction partner which results in $h_{j \rightarrow n} = h_{j \rightarrow i}, \ \forall n \in \{1, .., N\}$ for any particular $i$ and $j$ in each layer.

**Definition 3.2** (Dynamic relational learning). For any valid pair of entities $(x_i, x_j)$, the relational learning process $f$ is *dynamic* if representation of $x_j$ can dynamically adapt for $x_i$ based on some computational or optimization criteria.

Formally, for any pair $(x_i, x_j)$, their representations in dynamic relational learning are:

$$h_{i \rightarrow j} = \psi(x_i; \theta_i), \quad h_{j \rightarrow i} = \psi(x_j; \theta_j, \phi_{ij})$$

where $\phi_{ij}$ explicitly conditions the representation of $x_j$ on pair-specific $(x_i, x_j)$ dynamics and $\theta_j$ represents pair-invariant parameters for $x_j$. Thus, under dynamic relational learning, $h_{j \rightarrow n}$ is not constrained to be equal to $h_{j \rightarrow i}, \ \forall n \in \{1, ..., N\}, n \neq i$.

With the following characterizations, we can show that attention mechanism in transformers perform static relational learning.

**Proposition 3.3.** *Standard attention in transformers implements static relational learning.*

*Proof.* In standard attention, relational modeling between any two tokens $(x_i, x_j)$ can be defined as:

$$f(x_i, x_j) = \alpha_{ii} \cdot h_{i \rightarrow j} + \alpha_{ij} \cdot h_{j \rightarrow i}$$
$$\alpha_{ii} = e(h_{i \rightarrow j}, h_{i \rightarrow j}), \quad \alpha_{ij} = e(h_{i \rightarrow j}, h_{j \rightarrow i})$$

where $e(\cdot) \rightarrow \mathbb{R}$ is the (possibly learnable) attention scoring function typically defined as an inner product. Representations $h_{i \rightarrow j}$ and $h_{j \rightarrow i}$ are computed as follows:

$$h_{i \rightarrow j} = x_i W_q$$
$$h_{j \rightarrow i} = x_j W_{kv}$$

where $W_q, W_{kv} \in \mathbb{R}^{d \times d_{model}}$ are learned projection matrices. Then, entity representations for relational learning $f(x_u, x_j)$ for any $u \neq i$ is formulated as:

$$h_{u \rightarrow j} = x_u W_q$$
$$h_{j \rightarrow u} = x_j W_{kv}$$

where $h_{j \rightarrow u} = h_{j \rightarrow i} = x_j W_{kv}, \ \forall u \in \{1, ..., N\}$. Therefore, standard attention computes static relational learning. *Note: We simplify the key-value projections for notational clarity and to visually emphasize pair-wise dynamics.*

As a consequence of Proposition 3.3, no amount of *sharpening* of attention coefficients $\alpha$ can alleviate the static nature

of relational learning in standard attention. This also extends to other forms of possibly multi-dimensional relational information such as edge-features in some GNNs as long as the underlying representations that formulate such relational information are static.

# 4. Attention with Dynamic Relational Priming

Having established our categorization of relational learning, we now introduce our proposed methodology. We first show how standard attention is modified to enable prime attention. Next, we discuss how the learnable primers are constructed to enable pair-specific modulations. Finally, we show how prime attention enables dynamic relational learning to overcome the limitations of standard attention in MTS.

**Prime Attention for Transformers** Mathematically, given tokens $x_i, x_j \in \mathbb{R}^{d_{token}}$ for some $i, j \in \{1, ..., N\}$ where $N$ denotes the number of tokens, the query vector $q_i = x_i W_q$ and key, value vectors $k_j = x_j W_k$, $v_j = x_j W_v$ are derived with projection matrices $W_q, W_k, W_v \in \mathbb{R}^{d_{token} \times d_{model}}$. Then, standard attention computation for token $x_i$ is defined as:

$$x_i' = \sum_{j=1}^{N} softmax_j(e(q_i, k_j))v_j$$

where $e(\cdot)$ represents some pair-wise scoring function such as an inner product. We have established in Proposition 3.3 how this formulation results in static relational learning.

On the other hand, prime attention enables each token to present dynamically modulated perspective of itself based on the specific token-pair context. This is achieved as:

$$\widetilde{x}_i' = \sum_{j=1}^{N} softmax_j(e(q_i, \widetilde{k}_j))\widetilde{v}_j$$

where $\widetilde{k}_j$ and $\widetilde{v}_j$ are defined as:

$$\widetilde{k}_j = k_j \odot \mathcal{F}_{ij}$$
$$\widetilde{v}_j = v_j \odot \mathcal{F}_{ij}$$

where $\odot$ represents element-wise multiplication and $\mathcal{F}_{ij} \in \mathbb{R}^{d_{model}}$ represents a learnable primer designed to give pair-wise relational context to token-pair $x_i$ and $x_j$. As such, $\widetilde{k}_j$ and $\widetilde{v}_j$ are updated versions of $k_j$ and $v_j$ modulated specifically to maximize relevant information extraction from that particular token-pair ($x_i$-$x_j$). For example, $\mathcal{F}_{ij}$ might modulate $k_j$ and $v_j$ to make lead-lag correlations more easily

discoverable if the model learns that such relationship exists between tokens $i$ and $j$. For a different pair $x_k$ and $x_j$, $\mathcal{F}_{kj}$ may highlight certain parts of $k_j$ and $v_j$ to make instantaneous couplings more easily discoverable instead.

**Learnable Modulator Generation** The primer $\mathcal{F}_{ij}$ can be learned from scratch (with random initialization near the identity $I$) or expanded, in a learnable fashion, from estimated known patterns that exist in the domain (like lead-lag relationships and other temporal dynamics). In the simplest case, $\mathcal{F}_{ij}$ can be constructed from some random initialization $s \sim \mathcal{N}_{d_{model}}(\mu, \Sigma)$ and made learnable through a neural network, such as $\mathcal{F}_{ij} = MLP(s_{ij})$.

To potentially accelerate the discovery of patterns that are known to exist in the given data, $s$ can be initialized with estimations of such known patterns. An example of such pattern in MTS is lead-lag correlations that exist between channels. Following from Zhao & Shen (2024), lead-lag coefficients between channels $x_i$ and $x_j$ for all possible leading steps $\{1, ..., \mathcal{L}\}$ within a look-back window $L = \{t - \mathcal{L} + 1 : t\}$ can be calculated as follows:

$$\{R_{i,j}^{(t)}(\tau)\}_{\tau=1}^{\mathcal{L}} = \frac{1}{\mathcal{L}} FFT^{-1}(FFT(x_j^L) \odot FFT_{conj}(x_i^L))$$

where $FFT(\cdot)$ denotes the Fast Fourier Transform and $x_i^L$ denotes the look-back window $L$ of channel $i$.

Then, the initialization can be biased with $\widetilde{s_{ij}} = \sigma(R_{i,j}^{(t)})W$ where $W \in \mathbb{R}^{\mathcal{L} \times d_{model}}$ represents some learnable projection matrix and the optional $\sigma(\cdot)$ can be a smoothing function such as $tanh(\cdot)$ used to normalize and smooth-out the raw values of $R_{i,j}^{(t)}$. The filter $\mathcal{F}_{ij} = MLP(\widetilde{s_{ij}})$ can then learn to selectively amplify or dampen the introduced pattern information through the learnable neural network, $MLP(\cdot)$. While lead-lag is a one example of a relational pattern that may exist in MTS data, other information (such as instantaneous correlation, non-linear coupling, etc.) can also be used in conjunction to bias the primer. The primer itself remains learnable and adaptive despite this initialization bias and can learn more complex patterns on top of the initialization bias or learn entirely new patterns instead.

**Dynamic Relational Learning with Prime Attention** Here we show how prime attention enables dynamic relational learning, providing the representational adaptability crucial for capturing heterogeneous relational patterns characteristic of MTS systems.

**Proposition 4.1.** *Prime attention enables dynamic relational learning in transformers.*

*Proof.* In prime attention, relational modeling between any two entities $(x_i, x_j)$ can be defined as:

$$f(x_i, x_j) = \alpha_{ii} \cdot h_{i \to j} + \alpha_{ij} \cdot \widetilde{h}_{j \to i}$$
$$\alpha_{ii} = e(h_{i \to j}, h_{i \to j}), \quad \alpha_{ij} = e(h_{i \to j}, \widetilde{h}_{j \to i})$$

where entity representation $h_{i \to j}$ and $\widetilde{h}_{j \to i}$ are derived as:

$$\widetilde{h}_{j \to i} = h_{j \to i} \odot \mathcal{F}_{ji}$$
$$h_{i \to j} = x_i W_q, \quad h_{j \to i} = x_j W_{kv}$$

where $W_q, W_{kv} \in \mathbb{R}^{d \times d_{model}}$ are learned projection matrices. Then, relational learning $f(x_v, x_j)$ for any $v \neq i$ is formulated as:

$$f(x_v, x_j) = \alpha_{vv} \cdot h_{v \to j} + \alpha_{vj} \cdot \widetilde{h}_{j \to v}$$

and it can be seen that:

$$\widetilde{h}_{j \to i} = x_j W_{kv} \odot \mathcal{F}_{ji}$$
$$\widetilde{h}_{j \to v} = x_j W_{kv} \odot \mathcal{F}_{jv}$$

and since $\mathcal{F}_{ji}$ and $\mathcal{F}_{jv}$ are independent learnable parameters, they are not constrained to be equal. In general, after training, $\mathcal{F}_{ji} \neq \mathcal{F}_{jv}$ for $v \neq i$. As a result, $\widetilde{h}_{j \to v} \neq \widetilde{h}_{j \to i}$ for $v \neq i$. Therefore, prime attention performs dynamic relational learning where entities can dynamically adapt to each pair-wise interaction.

As a consequence of Proposition 4.1, the representational plasticity of prime attention's dynamic relational learning allows each token to present computationally optimized perspectives for different interaction partners. In domains like MTS where there can be high degrees of relational heterogeneity between different pair-wise interactions, prime attention's dynamic relational learning can allow robust discovery of different patterns for different pair-wise relationships. In addition, we show that prime attention, much like standard attention, satisfies the universal approximation theorem in Section C and provide a gradient flow analysis for prime attention in Section D.

**Complexity Analysis** Overall, prime attention introduces controlled computational overhead while preserving asymptotic computational complexity of standard attention. The core additional cost stems from the learnable primer $\mathcal{F} \in \mathbb{R}^{N^2 \times d_{model}}$. While this approximately doubles the floating-point operations compared to standard attention, the overall computational complexity remains unchanged. In addition,

we show in Section 6.2 and Section E.2 how prime attention can use less memory and computation time compared to standard attention while achieving better performance. This is achieved primarily because prime attention can exhibit better performance using dramatically smaller input data size compared to standard attention, owing to its superior relational modeling capabilities. In our main experiments, we show that prime attention is a practical solution that can be seamlessly adopted to even high dimensional datasets.

**Implementation Details** The learned primers are calculated once before each transformer block and are applied in each attention layer. In our main experiments, we construct the primers by biasing their initialization with estimated lead-lag patterns as well as instantaneous correlations. Instantaneous correlations are calculated with a simple dot-product. Once initialized, the primers are then put through an MLP to make them fully adaptive and learnable. Due to shared parameters in the MLP, the actual parameter-count overhead from prime attention is negligible, adding around 1.5% on top of standard attention.

## 5. Prime Attention as Learnable Pair-wise Inductive Bias

Prime attention's dynamic relational learning mechanism can be seen as an instance of learnable inductive bias injected into every pair-wise relationship in attention mechanism. Viewing prime attention through this lens allows us to make strong justifications about why prime attention can enable superior relational modeling capabilities compared to standard attention despite standard attention's universal approximation guarantees.

While transformers with standard attention theoretically satisfy universal approximation guarantees (Cybenko, 1989; Yun et al., 2020; Kajitsuka & Sato, 2024; Hu et al., 2025) and can model any relationship pattern given sufficient depth, width, and training data, practical constraints make these theoretical capabilities largely inaccessible. In this constrained regime, appropriate inductive biases have shown success in directing model capacity toward learning meaningful patterns efficiently.

Inductive biases are prior assumptions a model holds that influence its ability to generalize to new, unseen data. Simply put, inductive biases serve to nudge the representation learning process towards directions where useful learning is more likely to occur. In graph representation learning, the presence of edges serves to inductively bias the model towards discovering topologically relevant patterns (Hamilton et al., 2017; Gilmer et al., 2017). In geometric GNNs, inductively biasing the model through invariant or equivariant computations have shown tremendous success in molecular modeling, allowing for faster model convergence while

*Table 1.* Comparison between standard attention and prime attention on recent state-of-the-art transformer models (abridged). Lower values indicate better performance. Full table is available in Section F, Table 9.

| Model Attn Type | | Timer-XL (2025) | | | | FreDF (2025) | | | | iTransformer (2024) | | | |
| --- | --- | --- | --- | --- | --- | --- | --- | --- | --- | --- | --- | --- | --- |
| | | Standard | | *Prime* | | Standard | | *Prime* | | Standard | | *Prime* | |
| | | MSE | MAE | MSE | MAE | MSE | MAE | MSE | MAE | MSE | MAE | MSE | MAE |
| ETTh1 | Avg. | 0.453 | 0.445 | **0.438** | **0.435** | 0.452 | 0.443 | **0.449** | **0.441** | 0.449 | 0.443 | **0.440** | **0.438** |
| ETTh2 | Avg. | 0.396 | 0.415 | **0.389** | 0.412 | 0.390 | 0.411 | **0.389** | **0.409** | 0.385 | 0.409 | **0.377** | **0.404** |
| ETTm1 | Avg. | 0.441 | 0.428 | **0.427** | **0.420** | 0.402 | 0.402 | **0.400** | 0.402 | 0.405 | 0.409 | **0.394** | **0.404** |
| ETTm2 | Avg. | 0.293 | 0.334 | **0.290** | **0.332** | 0.289 | 0.327 | 0.289 | 0.327 | 0.287 | 0.331 | **0.282** | **0.327** |
| Weather | Avg. | 0.277 | 0.311 | **0.259** | **0.282** | 0.262 | 0.279 | **0.256** | **0.276** | 0.261 | 0.281 | **0.254** | **0.277** |
| Solar-Energy | Avg. | 0.282 | 0.284 | **0.279** | 0.285 | 0.234 | 0.254 | **0.229** | **0.251** | 0.238 | 0.265 | **0.229** | **0.258** |
| ECL | Avg. | 0.179 | 0.273 | **0.176** | **0.268** | 0.169 | 0.260 | **0.168** | **0.258** | 0.174 | 0.267 | 0.175 | **0.266** |
| Traffic | Avg. | 0.443 | 0.288 | **0.436** | 0.289 | 0.428 | 0.276 | **0.424** | **0.274** | 0.430 | 0.282 | **0.426** | **0.280** |
| PEMS03 | Avg. | - | - | - | - | 0.138 | 0.242 | **0.133** | **0.237** | 0.142 | 0.248 | **0.139** | **0.243** |
| PEMS08 | Avg. | - | - | - | - | 0.172 | 0.250 | **0.161** | **0.241** | 0.183 | 0.263 | **0.167** | **0.248** |

using dramatically fewer data samples during training (Vignac et al., 2020; Joshi et al., 2023; Batzner et al., 2022). Even positional encoding in transformers (Vaswani et al., 2017; Su et al., 2024; Barbero et al., 2025) can be seen as inductively biasing the model with explicit assumptions of sequential structure and distance information of tokens. Overall, inductive biases typically don't expand what the model can theoretically represent. However, by biasing the model through assumptions and contextually relevant information, it can dramatically improve what the model can learn efficiently with limited data and model capacity.

By inductively biasing pair-wise relationships, prime attention can enable faster and more reliable discoveries of different patterns that exist between channels. This can be particularly beneficial in MTS systems with high degrees of channel heterogeneity. Prime attention's ability to achieve comparable performance while using only a fraction of the sequence length used in standard attention (as shown in Section 6.2) aligns well with observed benefits of inductive bias in other domains and present notable positive implications for MTS modeling in data-scarce systems.

## 6. Experiments

**Experimental Setup** We use 11 widely-acknowledged benchmark datasets in our experiments made up of 9 long-term and 2 short-term datasets. These include Weather (Wu et al., 2021), Solar (Lai et al., 2018), Traffic (Wu et al., 2021) and more. More information on the datasets can be found in Section A.1 and experimental details can be found in Section A.2.

### 6.1. Forecasting Evaluation

For the main forecasting benchmarks, we fix the look-back window length at $L = 96$ and predict on horizon values $H \in \{96, 192, 336, 720\}$ except for the PEMS data where $H \in \{12, 24, 48, 96\}$. We also provide the averaged performance across all horizons. For any entry that we could not

get reliable results for (due to the official codebase being unavailable, having reproducibility issues, etc.), we simply mark them with the - symbol.

We use 3 state-of-the-art MTS transformers–Timer-XL (Liu et al., 2025b), FreDF (Wang et al., 2025), and iTransformer (Liu et al., 2024)–as backbone architectures to compare prime and standard attention in MTS forecasting. The prime attention version of these transformers are obtained by simply swapping out the main attention block with prime attention changing nothing else and keeping with the original hyper-parameters, re-tuning only dropout. The results are shown in Table 1. For each architecture, forecasting accuracy of standard and prime attention are recorded side-by-side with better performance in each model colored in **teal** if prime attention is better and in **brown** if standard attention is better. We use the abridged version here and the full table is available in Section F, Table 9. We observe consistent performance improvements using prime attention across datasets and models. On datasets with heterogeneous channels such as Solar and Weather where different channels record different physical properties, we observe average improvement across models of up to 4.0% when using prime attention. Specifically, prime attention shows a 6.5% improvement against standard attention on Weather dataset using Timer-XL. On the other hand, on ECL and Traffic where all channels record the same attribute (i.e. electric consumption and traffic flow, respectively), prime attention's performance gains were marginal. This aligns well with our hypothesized benefits of prime attention and show that in cases where channel relationships are relatively homogenous, standard attention is still capable of robust performance. On short-term forecasts, prime attention achieved an average performance gain of 5.5%, indicating its ability to accurately capture noisy short-term patterns that standard attention struggles with.

In addition, we also compare prime attention against a selection of MTS architectures that perform sophisticated relational learning beyond standard attention. These include

*Table 2.* Performance comparison between prime attention and existing works in MTS that enhance and leverage relational information (abridged). Lower values indicate better performance. Full table is available in Section F, Table 8.

| Model Metrics | | *Prime Attn.* (Ours) MSE | MAE | DIFF Transformer (2025) MSE | MAE | LagTS (2025) MSE | MAE | LIFT (2024) MSE | MAE | Autoformer (2021) MSE | MAE |
|---|---|---|---|---|---|---|---|---|---|---|---|
| ETTh1 | Avg. | 0.440 | **0.438** | 0.446 | 0.440 | **0.439** | 0.439 | 0.443 | 0.439 | 0.496 | 0.487 |
| ETTh2 | Avg. | **0.377** | **0.404** | 0.386 | 0.411 | 0.382 | 0.405 | 0.386 | 0.408 | 0.450 | 0.459 |
| Exchange | Avg. | **0.339** | **0.392** | 0.357 | 0.401 | 0.341 | 0.397 | 0.356 | 0.403 | 0.613 | 0.539 |
| Weather | Avg. | **0.254** | **0.277** | 0.259 | 0.280 | 0.258 | 0.281 | 0.260 | 0.282 | 0.338 | 0.382 |
| ECL | Avg. | 0.175 | 0.266 | **0.173** | **0.265** | - | - | 0.193 | 0.278 | 0.227 | 0.338 |
| Traffic | Avg. | **0.426** | **0.280** | 0.429 | 0.281 | - | - | 0.513 | 0.325 | 0.628 | 0.379 |

LagTS (Liu et al., 2025a) and LIFT (Zhao & Shen, 2024) as well as our implementation of the differential transformer (DIFF transformer) (Ye et al., 2025) using iTransformer (Liu et al., 2024) as backbone. We also include Autoformer (Wu et al., 2021), an earlier but seminal work in MTS transformers that replaces standard attention with an auto-correlation mechanism for better pattern discovery. Their abridged results are reported in Table 2, where prime attention continues to exhibit superior performance in the majority of instances. The best results in each dataset are shown in teal. The unabridged, full table is available in Section F, Table 8.

### 6.2. Sequence Length Efficiency Analysis

Here, we perform a sequence length (or look-back window length $L$) analysis for standard and prime attention. Namely, we are interested in seeing how performance improves with longer sequence length (or more input data). Performance for each model is measured across $L \in \{16, 32, 48, 64, 80, 96\}$ and the prediction horizon is fixed at $H = 96$. We show the results in Figure 1, using iTransformer (Liu et al., 2024) as backbone for both standard and prime attention.

Remarkably, prime attention not only shows improved performance at each $L$ as expected, but it also shows that it can match or slightly outperform standard attention's performance with significantly less input sequence length in some instances. To illustrate this, we use a horizontal green dotted line labeled *target threshold* to mark the performance of standard attention when using $L = 96$. As shown in Figures 1a and 1b, prime attention breaks through the target threshold using sequence lengths of 48 and 64, which translates to using 40% and 33% less input data compared to standard attention, respectively. We show in Section E.2 how this can translate to prime attention exhibiting lower *effective* complexity than standard attention, using less memory and lower computation time than standard attention while achieving superior performance. While such significant results were not observed across all datasets, it does align well with the observed benefits of inductive biases in other domains. It also strengthens our argument from Section 5 that prime attention acts as a learnable inductive bias in attention to enable and accelerate effective discoveries of unique rela-

tional dynamics across pair-wise interactions. In addition, this finding may be of interest in domains where input data is scarce. We provide additional results in Section E.4.

### 6.3. Attention Map Analysis

We perform attention map analysis to compare standard and prime attention. As shown in Figure 2, standard attention (on the left) focuses disproportionate attention on one particular channel (shown as yellow vertical line) for the Solar dataset. While this is not a problem in isolation, the fact that all channels develop an over-reliance on a single channel while largely ignoring their own signals can potentially indicate that the model is converging to an easy solution and not necessarily a good one. In addition, over-reliance on a single channel can be problematic in deployed systems as it presents a single point of failure (SPOF) for the learnability of the model. Prime attention (displayed on the right) on the other hand completely remedies this issue and learns to prioritize self-connections (along the diagonal). This may stems from prime attention being equipped to learn fundamentally better pair-wise patterns than standard attention, allowing the model to focus on and learn critical self-signals instead of prematurely converging to focus on a single channel, though this remains a speculative analysis. In this case, prime attention's superior relational modeling abilities enable the model to actually learn to ignore spurious inter-channel dependencies, acting like a *soft* CI model instead. Prime attention's performance improvements of 3.8% over standard attention verifies the validity of this approach on the Solar dataset. We find that prime attention generally learns complex and diverse patterns, showing that it is also capable of dampening self-signals instead of prioritizing them in other datasets as further illustrated in Section E.3.

In the case of the ECL dataset where both standard and prime attention show comparable performance (shown in Table 1), we see that their attention maps show different patterns, as illustrated in Figure 3. Prime attention learns to focus on a substantially larger number of channels compared to standard attention, showing more diverse learning patterns. This may also translate to better reliability from a SPOF perspective.

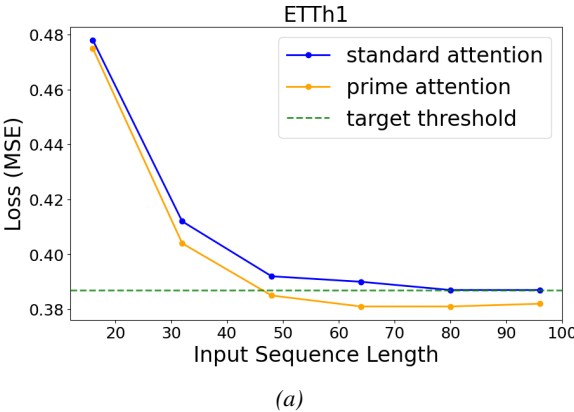 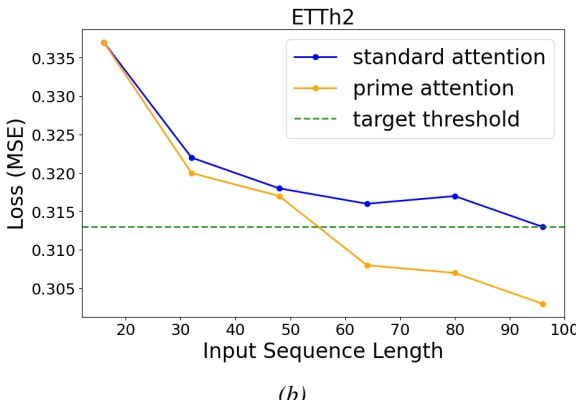

*(a)* *(b)*

*Figure 1.* Comparing performance between standard (blue) and prime (yellow) attention at various input sequence length (or look-back window $L$). The horizontal target threshold line (green, dotted) represents standard attention's performance at $L = 96$. Lower values indicate better performance.

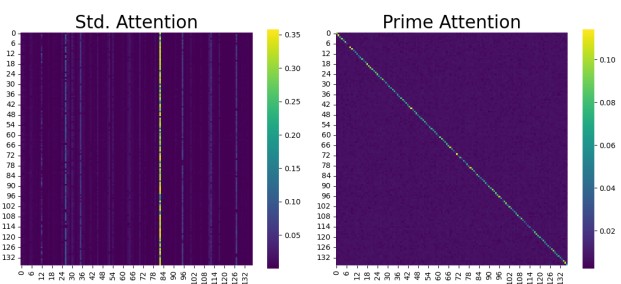

*Figure 2.* Attention map comparison between standard and prime attention on the Solar dataset. Standard attention (left) shows an over-reliance on a single channel (displayed as a yellow vertical line) which may indicate that the model is prematurely converging to an easy solution, unable to capture more diverse and complex patterns. Prime attention (right) on the other hand shows its ability to act as a *soft* CI model, learning to prioritize self-signals along the diagonal instead of developing an over-reliance on a single channel.

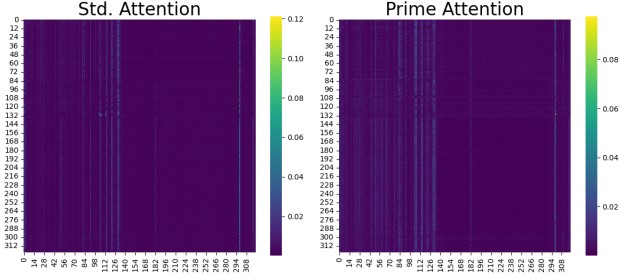

*Figure 3.* Attention map comparison between standard (left) and prime (right) attention on the ECL dataset. Despite being comparable in performance on this dataset, prime attention exhibits a more diverse attention pattern than standard attention, learning to focus on a larger number of channels compared to standard attention. This can be seen from the larger number of vertical lines present in prime attention's attention map.

Overall, we note that attention map analysis shows correlational, rather than causal, evidence and that this should not be used in isolation to draw conclusions about prime attention's performance. Rather, it should be understood within the broader context of quantitative evaluations in Section 6.1 and Section 6.2.

### 6.4. Parameter Effectiveness Analysis

Standard multi-headed attention itself can be interpreted as a mechanism for capturing multiple, parallel, and dynamic patterns. Since prime attention's primers can be seen as introducing extra information on top of the existing attention heads, we equip standard attention with an additional attention head to compare its performance with prime attention. For this, we increase standard attention's model dimensions proportionately to keep per-head dimensions consistent. This ends up increasing standard attention's parameter count by around 38%, orders of magnitude higher than prime attention. We find in Table 3 that naively increasing parameter count does not translate to better performance in standard attention even when it has substantially higher parameter count than prime attention. This further demonstrates that prime attention enables fundamentally better learning of inter-channel relationships and that naively scaling up standard attention does not guarantee such patterns will be adequately learned by standard attention.

*Table 3.* Parameter effectiveness analysis. Best result (in MSE) in each row is shown in **bold**.

| Dataset | Std. Attn. (8 heads) | Std. Attn. (9 heads) | Prime Attn. (8 heads) |
|---|---|---|---|
| ETTh1 | 0.386 | 0.385 | **0.378** |
| Weather | 0.176 | 0.176 | **0.170** |
| Solar | 0.207 | 0.209 | **0.199** |

## 6.5. Additional Analysis

We perform ablation studies in Section E.1 to study the effects of sparsity and initialization strategies in designing the learnable modulator $\mathcal{F}$. Complexity analysis is done in Section E.2 where we show that prime attention can exhibit lower *effective* memory and computation time than standard attention. We also experiment modulating only $K$ and only $V$ in Section E.5 and show that modulating both $K$ and $V$ as we do in our methodology make sense empirically as well as intuitively.

## 7. Conclusion

We introduced prime attention, a novel attention mechanism enabling dynamic relational learning through pair-wise token modulation for multivariate time series (MTS) forecasting. Unlike standard attention's static relational learning, prime attention dynamically adapts token representations for each specific interaction via learnable primers, better capturing the heterogeneous inter-channel de-pendencies characteristic of many MTS systems. Our results showed that prime attention consistently outperforms standard attention and demonstrate the practical benefits of dynamic relational learning in MTS.

## Impact Statement

This paper presents work whose goal is to advance the field of Machine Learning in the domain of multivariate time series. There are many potential societal consequences of our work, none of which we feel must be specifically highlighted here.

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

# A. Implementation Details

## A.1. Dataset Information

We provide additional information on the real-world datasets used in our experiments. ECL, ETT (4 variations), Traffic, and Weather datasets are from (Wu et al., 2021). Solar dataset is from (Lai et al., 2018). We use dataset information and descriptions from (Liu et al., 2024; Wang et al., 2024c; Zhang & Yan, 2023).

1. **ETTh1 & ETTh2** contain 7 indicators of an electricity transformer in two years, including oil temperature, load characteristics, etc. Data points are recorded every hour.

2. **ETTm1 & ETTm2** contain the same data as with the ETTh datasets but recorded over 15 minute intervals.

3. **Exchange** contains daily exchange rates data from 8 countries between 1990 and 2016.

4. **Weather** contains 21 meteorological indicators in the U.S over the entire year of 2020, including features like visibility, wind speed, etc.

5. **Solar** records solar power production of 137 PV plants in 2006, sampled every 10 minutes.

6. **ECL** records the hourly electricity consumption data of 321 clients.

7. **Traffic** records road occupancy rates measured by 862 sensors in San Francisco freeways from 2015 to 2016.

8. **PEMS (03, 08)** contain traffic flow data in California at 5 minutes intervals.

*Table 4.* Details of the datasets used in our experiments. The ETT datasets used training, validation, and testing split of 6:2:2 while the rest of the datasets used a 7:1:2 splits, in line with the experimental setups from (Liu et al., 2024; Zhang & Yan, 2023) and other baselines.

| Dataset | Dim | Prediction Length | Dataset Size | Frequency | Information |
|---|---|---|---|---|---|
| ETTh1 | 7 | {96, 192, 336, 720} | (8545, 2881, 2881) | Hourly | Electricity |
| ETTh2 | 7 | {96, 192, 336, 720} | (8545, 2881, 2881) | Hourly | Electricity |
| ETTm1 | 7 | {96, 192, 336, 720} | (34465, 11521, 11521) | 15min | Electricity |
| ETTm2 | 7 | {96, 192, 336, 720} | (34465, 11521, 11521) | 15min | Electricity |
| Exchange | 8 | {96, 192, 336, 720} | (5120, 665, 1422) | Daily | Economy |
| Weather | 21 | {96, 192, 336, 720} | (36792, 5271, 10540) | 10min | Weather |
| ECL | 321 | {96, 192, 336, 720} | (18317, 2633, 5261) | Hourly | Electricity |
| Traffic | 862 | {96, 192, 336, 720} | (12185, 1757, 3509) | Hourly | Transportation |
| Solar-Energy | 137 | {96, 192, 336, 720} | (36601, 5161, 10417) | 10min | Energy |
| PEMS03 | 358 | {12, 24, 48, 96} | (15617, 5135, 5135) | 5min | Transportation |
| PEMS08 | 170 | {12, 24, 48, 96} | (10690, 3548, 3548) | 5min | Transportation |

## A.2. Experimental Details

All experiments are implemented in PyTorch (Paszke et al., 2019) and run on a single NVIDIA A100 80GB GPU. We use the ADAM (Kingma & Ba, 2014) optimizer and MSE loss function for all training. We use averaged results from 5 runs with random seeds. For more information on dataset-specific model configuration, refer to Table 5. For fair comparison against other transformer models, we follow the practices of recent state-of-the-art transformer baselines (Liu et al., 2024; 2025b; Zhang & Yan, 2023) and prepare our data with simple pre-processing steps and prediction block. Concretely, this means that we only use a linear layer for tokenization and do not make any use of more complicated signal processing techniques or data enriching strategies to enhance the model beyond its core technical contribution. For prediction, this also means that we are only using a linear layer as decoder for forecasting. One notable exception is reverse instance normalization, a widely adopted technique that is utilized in nearly every modern MTS architecture and is discussed further in Section A.4. These steps are taken strictly for benchmarking purposes and more sophisticated data engineering and forecasting methodologies should be integrated as seen fit in practical applications.

*Table 5.* Model configurations across different datasets with iTransformer (Liu et al., 2024) backbone. We keep the hyperparameters used with standard attention (with dropout being the only exception) in prime attention to ensure fair comparison.

| Dataset | Model Hyper-Parameter | | | | Training Process | | |
|---------|:---------:|:------:|:-------:|:-------------:|:----:|:----------:|:------:|
|         | Model Dim | Layers | Dropout | Learning Rate | Loss | Batch Size | Epochs |
| ETTh1 | 256 | 2 | 0.1 | 0.0001 | MSE | 128 | 10 |
| ETTh2 | 128 | 2 | 0.1 | 0.0001 | MSE | 128 | 10 |
| ETTm1 | 128 | 2 | 0.2 | 0.0001 | MSE | 128 | 10 |
| ETTm2 | 128 | 2 | 0.8 | 0.0001 | MSE | 128 | 10 |
| Weather | 128 | 3 | 0.4 | 0.0005 | MSE | 128 | 10 |
| Electricity (ECL) | 512 | 3 | 0.2 | 0.0005 | MSE | 4 | 10 |
| Solar | 128 | 6 | 0.2 | 0.0001 | MSE | 4 | 10 |
| Traffic | 128 | 4 | 0.2 | 0.001 | MSE | 2 | 10 |
| Exchange | 64 | 2 | 0.8 | 0.0001 | MSE | 128 | 10 |
| PEMS03 | 512 | 4 | 0.1 | 0.0001 | MSE | 2 | 10 |
| PEMS08 | 512 | 2 | 0.1 | 0.0001 | MSE | 4 | 10 |

## A.3. Metrics Used

For metrics used in our benchmarking, we utilize both mean squared error (MSE) and mean absolute error (MAE). MSE penalizes larger errors more heavily due to its quadratic nature, making it particularly sensitive to outliers. This characteristic is valuable for applications where large deviations are especially problematic. Conversely, MAE treats all error magnitudes linearly, providing a more robust measure of average model performance that is less influenced by occasional extreme predictions. By reporting both metrics, we offer a more comprehensive evaluation of model performance: MSE highlights models that avoid significant errors, while MAE better reflects the typical prediction accuracy a user might expect in practice.

$$MSE(y, \hat{y}) = \frac{1}{n} \sum_{i=1}^{n} (y_i - \hat{y}i)^2 \tag{1}$$

$$MAE(y, \hat{y}) = \frac{1}{n} \sum_{i=1}^{n} |y_i - \hat{y}_i| \tag{2}$$

## A.4. Data Non-stationarity

MTS data exhibits pronounced non-stationarity characterized by persistent alterations in statistical attributes and joint distributions across time (Liu et al., 2022). To tackle this issue, reverse instance normalization (RevIN) (Kim et al., 2022) has been widely used in recent works, including all of the baseline transformers we are using to benchmark our proposed method. RevIN addresses distribution shift in MTS data by first normalizing input data using instance-specific statistics and then de-normalizing the model output to restore the original distribution. More formally,

$$\text{Normalization:} \quad \hat{x} = \gamma \cdot \frac{x - \mu_x}{\sigma_x} + \beta \tag{3}$$

$$\text{Denormalization:} \quad \hat{y} = \sigma_x \cdot \tilde{y} + \mu_x \tag{4}$$

where $\mu_x$ and $\sigma_x$ are the instance-specific mean and standard deviation, while $\gamma$ and $\beta$ are learnable parameters.

*Table 6.* Averaged forecasting results from 5 independent runs with standard deviation.

| Horizon | ETTh1 | ETTh2 | Weather | ECL | Traffic |
|---|---|---|---|---|---|
| 96 | $0.378 \pm 0.001$ | $0.297 \pm 0.001$ | $0.170 \pm 0.001$ | $0.146 \pm 0.0005$ | $0.395 \pm 0.0001$ |
| 192 | $0.432 \pm 0.001$ | $0.376 \pm 0.001$ | $0.218 \pm 0.001$ | $0.164 \pm 0.001$ | $0.417 \pm 0.0002$ |
| 336 | $0.473 \pm 0.001$ | $0.413 \pm 0.001$ | $0.275 \pm 0.001$ | $0.179 \pm 0.001$ | $0.431 \pm 0.0001$ |
| 720 | $0.479 \pm 0.001$ | $0.420 \pm 0.0008$ | $0.354 \pm 0.0005$ | $0.214 \pm 0.002$ | $0.463 \pm 0.001$ |
| Avg. | $0.441 \pm 0.001$ | $0.377 \pm 0.001$ | $0.254 \pm 0.0008$ | $0.176 \pm 0.001$ | $0.427 \pm 0.0003$ |

RevIN makes up one of the most widely used pre-processing techniques in transformers for MTS forecasting. For the most part, however, recent MTS transformer keep pre-processing and data engineering simple and straight-forward to ensure fair and consistent comparisons against other works for benchmarking purposes. We also follow this paradigm in our experiments.

### A.5. Stability of Main Results

We provide averaged results with standard deviation from 5 independent runs with random seeds to confirm the stability of our results in Table 6. We use iTransformer (Liu et al., 2024) as backbone here.

## B. Channel Strategies in MTS

Channel strategies in MTS are broadly divided into two categories: channel-independent (CI) (Zeng et al., 2023; Nie et al., 2022; Chen et al., 2025) and channel-dependent (CD) (Liu et al., 2024; Wang et al., 2024b;a). CI approaches model each channel independently without any cross-channel interactions, which bypasses the channel heterogeneity problem entirely. CI approaches have been shown to be highly robust but crucially lack capacity (Han et al., 2024) and generalizability (Qiu et al., 2025).

CD approaches, on the other hand, argue that certain patterns crucial for accurate MTS forecasting such as lagged information (Liu et al., 2025a; Zhao & Shen, 2024) or intricate correlations and physical laws (Chen et al., 2023; Yu et al., 2024) can only be captured by inter-channel modeling. Therefore, CD approaches allow cross-channel interactions to capture inter-channel relationships crucial for accurate forecasting. As a result, CD methods are susceptible to the potentially complex and heterogeneous inter-channel dynamics that, if not captured correctly, can lead to noise accumulation and performance degradation. Our proposed prime attention is a CD approach that explicitly tackles the channel-heterogeneity problem for transformers in MTS with the aim of making channel-wise attention more favorable in the domain of time series.

## C. Universal Approximation Theorem for Prime Attention

Standard attention mechanism in transformers have been proven to satisfy the universal approximation theorem. Given sufficient depth, width, and training data, standard attention can approximate any continuous sequence-to-sequence function to arbitrary precision (Cybenko, 1989; Yun et al., 2020). By showing that prime attention contains standard attention, we make a simple theoretical claim that prime attention also satisfies the universal approximation theorem and its guarantees.

**Theorem C.1.** *Prime attention satisfies the universal approximation theorem.*

*Proof.* We examine the function class relationship between prime attention and standard attention. Let $\mathcal{H}_{\text{standard}}$ denote the function class representable by standard attention and $\mathcal{H}_{\text{prime}}$ denote the function class representable by prime attention. Prime attention extends standard attention by introducing learnable element-wise modulation filters $\mathcal{F}_{ij} \in \mathbb{R}^{d_{\text{model}}}$ for each token pair $(x_i, x_j)$:

$$\widetilde{k}_j^{(i)} = k_j \odot \mathcal{F}_{ij}, \quad \widetilde{v}_j^{(i)} = v_j \odot \mathcal{F}_{ij} \tag{5}$$

We make the observation that prime attention strictly contains standard attention as a special case in its function class. Notably, when setting all modulation filters to the identity $\mathcal{F}_{ij} = 1_{d_{model}}, \ \forall i, j \in \{1, ..., N\}$, prime attention is reduced to:

$$\widetilde{k}_j^{(i)} = k_j \odot 1_{d_{model}} = k_j, \quad \widetilde{v}_j^{(i)} = v_j \odot 1_{d_{model}} = v_j \tag{6}$$

which equates to standard attention. Since $\mathcal{F}_{ij} \in \mathbb{R}^{d_{model}}$ and $1_{d_{model}} \subset \mathbb{R}^{d_{model}}$, we can show $\mathcal{H}_{standard} \subseteq \mathcal{H}_{prime}$. Therefore, prime attention can approximate *at least* all functions that standard attention can approximate. Since standard attention satisfies the universal approximation guarantees and can approximate any continuous sequence-to-sequence function $f : \mathbb{R}^{n \times d} \to \mathbb{R}^{m \times d}$ to arbitrary precision given sufficient resources, and since $\mathcal{H}_{standard} \subseteq \mathcal{H}_{prime}$, prime attention inherits the universal approximation guarantees of standard attention.

This theoretical guarantee ensures that prime attention does not sacrifice any representational capabilities of standard attention and that any function learnable by standard attention is also learnable by prime attention. As with standard attention's universal approximation guarantees, this assumes sufficient model depth and width as well as adequate training data and appropriate optimization strategy. In addition, proper initialization and training of learnable modulator $\mathcal{F}$ is also crucial.

## D. Gradient Flow Analysis for Prime Attention

We provide a gradient flow analysis comparing standard attention and prime attention. This analysis assumes a single layer with one attention head.

For both mechanisms, assume the input data $X \in \mathbb{R}^{N \times d_{model}}$ where $N$ is the number of tokens and $d_{model}$ is the model dimension. For standard attention,

$$Q = XW_Q, \quad K = XW_K, \quad V = XW_V \tag{7}$$

where $W_Q, W_K, W_V \in \mathbb{R}^{d_{model} \times d_{model}}$ are projection matrices.

**Standard Attention Gradient Flow**   The forward computation flow for standard attention is:

$$A = softmax(\frac{QK^T}{\sqrt{d_{model}}}) \tag{8}$$

$$O = AV \tag{9}$$

Given the gradient of $O$ as $\frac{\partial L}{\partial O}$ for standard attention, the gradients of parameters are formulated via:

$$\frac{\partial L}{W_V} = \frac{\partial L}{\partial O}\frac{\partial O}{\partial V}\frac{\partial V}{\partial W_V} \tag{10}$$

$$= X^T A^T (\frac{\partial L}{\partial O}) \tag{11}$$

$$\frac{\partial L}{\partial W_Q} = \frac{\partial L}{\partial O}\frac{\partial O}{\partial A}\frac{\partial A}{\partial Q}\frac{\partial Q}{\partial W_Q} \tag{12}$$

$$= \frac{1}{\sqrt{d_{model}}}X^T[A \odot (\frac{\partial L}{\partial O}V^T - A \odot (\frac{\partial L}{\partial O}V^T)J)]K \tag{13}$$

$$\frac{\partial L}{\partial W_K} = \frac{\partial L}{\partial O}\frac{\partial O}{\partial A}\frac{\partial A}{\partial K}\frac{\partial K}{\partial W_K} \tag{14}$$

$$= \frac{1}{\sqrt{d_{model}}}X^T[A \odot (\frac{\partial L}{\partial O}V^T - A \odot (\frac{\partial L}{\partial O}V^T)J)]Q \tag{15}$$

**Prime Attention Gradient Flow** Prime attention extends standard attention's computation with learnable pair-wise modulators that are applied to $K$ and $V$ as:

$$\widetilde{K} = K \odot \mathcal{F}, \quad \widetilde{V} = V \odot \mathcal{F} \tag{16}$$

where we assume the full modulator $\mathcal{F} \in \mathbb{R}^{N \times N \times d_{model}}$. Then, the forward flow for prime attention is:

$$\widetilde{A} = softmax(\frac{Q\widetilde{K}^T}{\sqrt{d_{model}}}) \tag{17}$$

$$\widetilde{O} = \widetilde{A}\widetilde{V} \tag{18}$$

Given the gradient of $\widetilde{O}$ as $\frac{\partial L}{\partial \widetilde{O}}$ for prime attention, the gradients of parameters are formulated via:

$$\frac{\partial L}{\partial W_V} = \frac{\partial L}{\partial \widetilde{O}} \frac{\partial \widetilde{O}}{\partial \widetilde{V}} \frac{\partial \widetilde{V}}{\partial V} \frac{\partial V}{\partial W_V} \tag{19}$$

$$= X^T (\widetilde{A}^T \odot \mathcal{F}^T) \frac{\partial L}{\partial \widetilde{O}} \tag{20}$$

$$\frac{\partial L}{\partial W_Q} = \frac{\partial L}{\partial \widetilde{O}} \frac{\partial \widetilde{O}}{\partial \widetilde{A}} \frac{\partial \widetilde{A}}{\partial Q} \frac{\partial Q}{\partial W_Q} \tag{21}$$

$$= (\frac{1}{\sqrt{d_{model}}}) X^T [\widetilde{A} \odot (\frac{\partial L}{\partial \widetilde{O}} \widetilde{V}^T - (\widetilde{A}^T \odot (\frac{\partial L}{\partial \widetilde{O}} \widetilde{V}^T)) J)] \widetilde{K} \tag{22}$$

$$\frac{\partial L}{\partial W_K} = \frac{\partial L}{\partial \widetilde{O}} \frac{\partial \widetilde{O}}{\partial \widetilde{A}} \frac{\partial \widetilde{A}}{\partial \widetilde{K}} \frac{\partial \widetilde{K}}{\partial K} \frac{\partial K}{\partial W_K} \tag{23}$$

$$= (\frac{1}{\sqrt{d_{model}}}) X^T ([\widetilde{A} \odot (\frac{\partial L}{\partial \widetilde{O}} \widetilde{V}^T - (\widetilde{A}^T \odot (\frac{\partial L}{\partial \widetilde{O}} \widetilde{V}^T)) J)]^T \odot \mathcal{F}^T) Q \tag{24}$$

and the gradient with respect to $\mathcal{F}$ across multiple paths is formulated as:

$$\frac{\partial L}{\partial \mathcal{F}_{path1}} = \frac{\partial L}{\partial \widetilde{O}} \frac{\partial \widetilde{O}}{\partial \widetilde{A}} \frac{\partial \widetilde{A}}{\partial \widetilde{K}} \frac{\partial \widetilde{K}}{\partial \mathcal{F}} \tag{25}$$

$$= (\frac{1}{\sqrt{d_{model}}}) K \odot [Q^T (\widetilde{A}^T \odot [\frac{\partial L}{\partial \widetilde{O}} \widetilde{V}^T - (\widetilde{A}^T \odot (\frac{\partial L}{\partial \widetilde{O}} \widetilde{V}^T)) J])] \tag{26}$$

$$\frac{\partial L}{\partial \mathcal{F}_{path2}} = \frac{\partial L}{\partial \widetilde{O}} \frac{\partial \widetilde{O}}{\partial \widetilde{V}} \frac{\partial \widetilde{V}}{\partial \mathcal{F}} \tag{27}$$

$$= V \odot [\widetilde{A}^T \frac{\partial L}{\partial \widetilde{O}}] \tag{28}$$

$$\frac{\partial L}{\partial \mathcal{F}} = \frac{\partial L}{\partial \mathcal{F}_{path1}} + \frac{\partial L}{\partial \mathcal{F}_{path2}} = \frac{\partial L}{\partial \widetilde{O}} \frac{\partial \widetilde{O}}{\partial \widetilde{A}} \frac{\partial \widetilde{A}}{\partial \widetilde{K}} \frac{\partial \widetilde{K}}{\partial \mathcal{F}} + \frac{\partial L}{\partial \widetilde{O}} \frac{\partial \widetilde{O}}{\partial \widetilde{V}} \frac{\partial \widetilde{V}}{\partial \mathcal{F}} \tag{29}$$

$$= (\frac{1}{\sqrt{d_{model}}}) K \odot [Q^T (\widetilde{A}^T \odot [\frac{\partial L}{\partial \widetilde{O}} \widetilde{V}^T - (\widetilde{A}^T \odot (\frac{\partial L}{\partial \widetilde{O}} \widetilde{V}^T)) J])] + V \odot [\widetilde{A}^T \frac{\partial L}{\partial \widetilde{O}}] \tag{30}$$

This gradient flow analysis shows how prime attention introduces controlled parameters through the dynamic modulator $\mathcal{F}$, which creates additional gradient pathways for learning pair-wise relational information while maintaining similar overall gradient flow structure as standard attention.

# E. Additional Analysis and Experiments

## E.1. Ablation Studies

**Sparsity Ablation**    We conduct an ablation study on the effect of sparsifying the pair-wise primer $\mathcal{F}$ such that only a certain percentage of token pairs are modulated with $\mathcal{F}$ at random and the rest effectively perform standard attention. The results are shown in Figure 4 where sparsity on the x-axis refers to the percentage of token-pairs not getting a pair-wise modulator.

This analysis reveals a nuanced relationship between the degree of pair-wise modulation and model performance. We find that 100% sparsity (effectively standard attention) always performs significantly worse than having even 20% of token pairs be assigned a pair-wise primer. However, we do not necessarily see a positively correlated relationship between pair-wise priming and performance and note that having less sparsity (and therefore more pair-wise primers) does not always result in superior performance. In fact, on some datasets, having some degree of sparsity in $\mathcal{F}$ slightly improved performance. Considering the memory requirements of prime attention, and our observation here that any amount of pair-wise modulation improves upon standard attention, it may be useful in practice to take sparsity as a hyperparameter and optimize for the highest degree of sparsity that still results in acceptable performance gains.

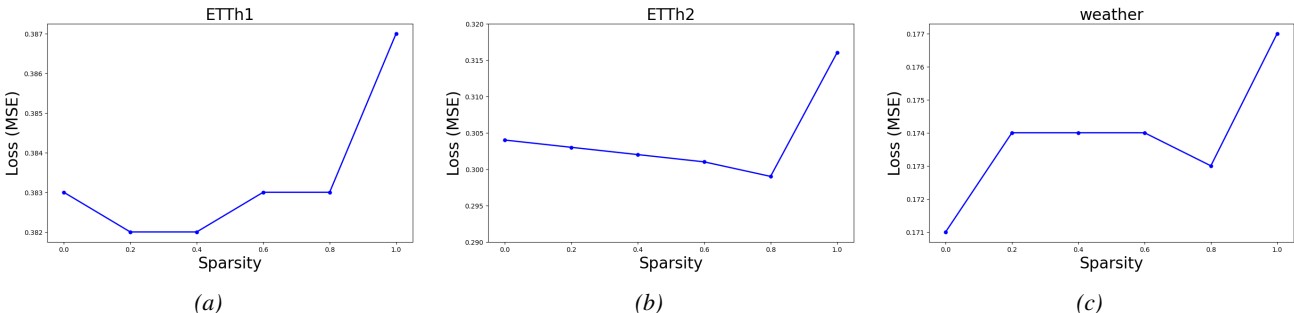

*Figure 4.* Ablation study on sparsity of pair-wise modulator $\mathcal{F}$. Sparsity refers to the amount (in percentage) of token pairs getting modulated (from left to right, 0% sparsity is full pair-wise modulation and 100% sparsity is equivalent to standard attention).

**Filter Ablation**    We study the effects of different initialization strategies for $\mathcal{F}$. We look at four initialization strategies and chart each of their performance across increasing prediction horizons $H \in \{96, 192, 336, 720\}$. Note that in all cases, the initialized modulator is made learnable with a 2-layer MLP. First, random initialization (in blue) uses Glorot initialization (Glorot & Bengio, 2010) centered around 1.0 to allow the model to effectively start at standard attention and learn relevant pair-wise relationships from scratch. We note that this generally performs worse than more sophisticated, domain appropriate strategies, as we show next. Second, we show initialization using estimated lead-lag information (in orange). This is mentioned in Section 4 and shows robust performance. Third, we initialize the modulator with instantaneous information and chart its performance in green. The estimated instantaneous correlation is calculated using a combination of Pearson correlation, rolling standard deviation, and rank-transformation information between channel pairs. We note that while this performs better than random initialization, it typically performs worse than lead-lag initialization. Lastly, we combine the lead-lag estimation and instantaneous correlations to create a full initialization (in red). The MLP takes both information and decides which information is more relevant for each channel-pair. Full initialization by far performs the best and shows stable performance and is used in our prime attention implementations.

Since these are merely initialization strategies, the MLP can ultimately learn to retain, modify, or even discard any information presented by more sophisticated initializations. Since MLPs satisfy the universal approximation theorem, given sufficient resources, the random initialization should be able to perform just as well as the full initialization across any given dataset. However, as discussed in Section 5, we find that injecting useful information to bias the model can make noticeable differences in practical scenarios.

## E.2. Complexity Analysis

We report empirical memory and time usage of prime attention using the Weather dataset and use standard attention as baseline. The results are shown in Figure 6. We use a Graph Attention Network (GAT) (Veličković et al., 2017; Brody et al., 2021) as the base architecture and systematically increase the density of the adjacency matrix to record memory and time

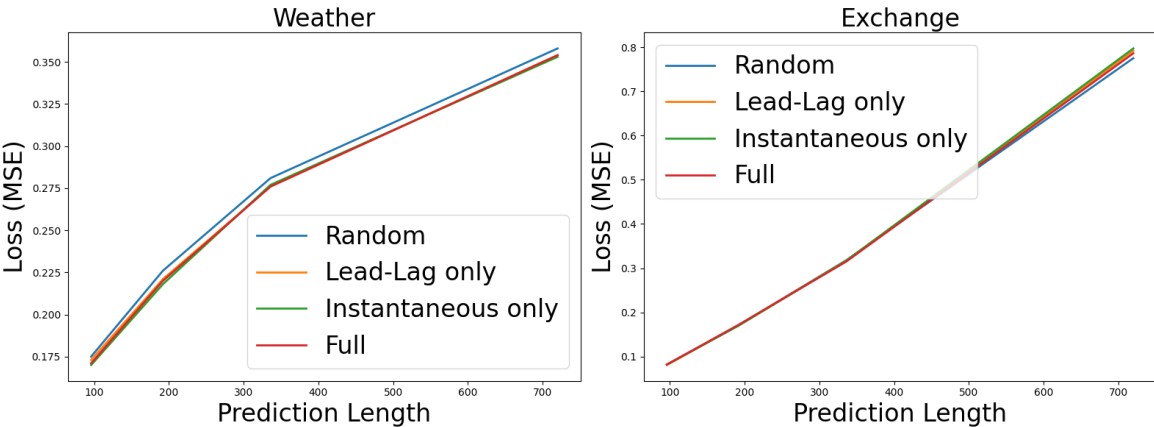

*Figure 5.* Effect that different initialization strategies for pair-wise modulation primer $\mathcal{F}$ has on performance (y-axis, lower is better) across prediction lengths (x-axis).

usage for both standard and prime attention. The density of the adjacency matrix refers to how many pair-wise connections are allowed as a percentage of a fully connected graph. In other words, higher adjacency matrix density indicates more pair-wise channel interactions and 100% density essentially becomes regular full, quadratic attention. In each case, a random graph is constructed based on the prescribed density.

While Figure 6 shows that prime attention indeed uses more memory and computation time than standard attention, the discrepancy is not so straightforward when considering performance (forecasting accuracy). To illustrate this point, we denote with red stars the points of equivalent performance of standard and prime attention. As it can be seen, prime attention using 60% adjacency matrix density matches standard attention's performance at 100% density. At 60% density, prime attention uses less memory and computation time than standard attention at 100% density (their gaps are shown as green triangles). This shows that prime attention is capable of having lower *effective* complexity than standard attention since prime attention using less data, memory, and computation time can, in certain instances, match or outperform standard attention.

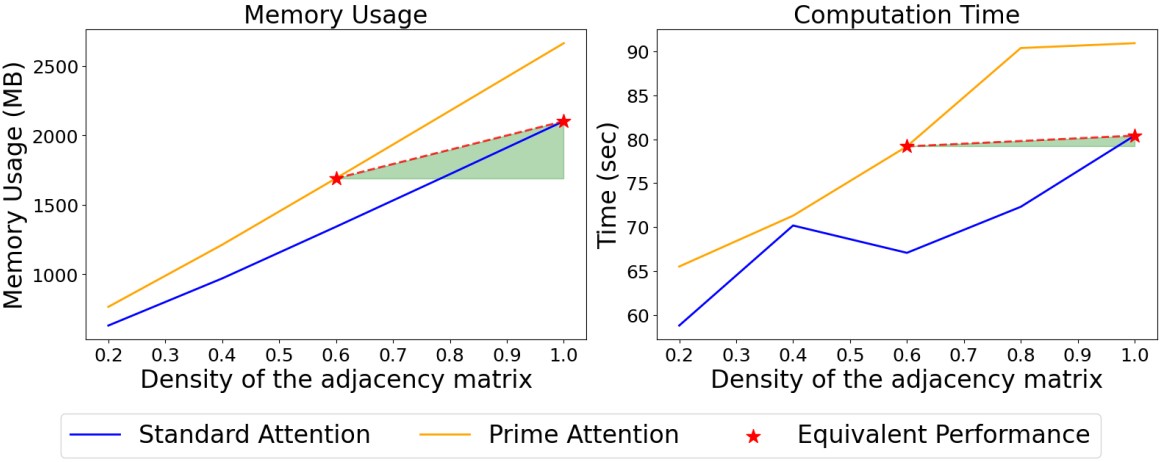

*Figure 6.* Analysis of memory (left) and computation time (right) for standard (blue) and prime (yellow) attention on the Weather dataset. The red star indicates observed equivalent performance (loss in MSE) between standard and prime attention.

### E.3. Attention Map Analysis

We perform attention map analysis for standard attention, prime attention, and their difference map to show the change from standard attention map to prime attention map. As shown in Figure 7, prime attention's attention coefficient distribution has similarities to standard attention but also notable differences as seen in their difference map. The difference map is a good

illustration of where standard and prime attention differ in their focus, with warmer colors (beige/red) indicating stronger focus from prime attention and colder colors (blue) indicating weaker focus from prime attention with respect to standard attention.

The attention maps of Solar dataset in Figure 8 demonstrates an interesting property of prime attention. As it can be seen, the standard attention map on the left shows disproportionate focus on one particular variable (shown as a thin yellow vertical line on the middle-right). While this is certainly not a problem in isolation, the fact that all channels have an over-reliance on a single particular channel while not valuing its own signals at all (along the diagonal) can potentially indicate that the model is converging to an easy solution and not necessarily a good one. In addition, an over-reliance on a single (or a small number of) signal(s) can be problematic as it can present a single point of failure in deployed systems whereby a loss of one sensor can devastate the performance of the entire system. Prime attention, on the other hand, completely remedies this issue and illustrates how the benefits of pair-wise modulation includes self-attention. Instead of putting an over-reliance on a single channel, prime attention enables effective pattern discovery for self-attention and learns to take the form of a *soft* CI model, whereby the model essentially learns to be channel-independent (given limitations of softmax). It's performance improvement of 3.8% from standard attention verifies the validity of this approach on this particular dataset.

In contrast to its ability to learn to be a soft CI model, Figure 9 demonstrates an instance where prime attention learns to do the opposite by learning to weaken its self-attention instead. As it can be seen from the difference map, prime attention actually weakens the contribution of self-attention of many channels in Figure 9, and instead strengthens a selection of cross-channel interactions.

Overall, attention map analysis for standard and prime attention on various datasets show that prime attention's relational modeling may be potent enough to act as a hybrid CD/CI model. While this is not the main focus of this work, it may merit further analysis for future work.

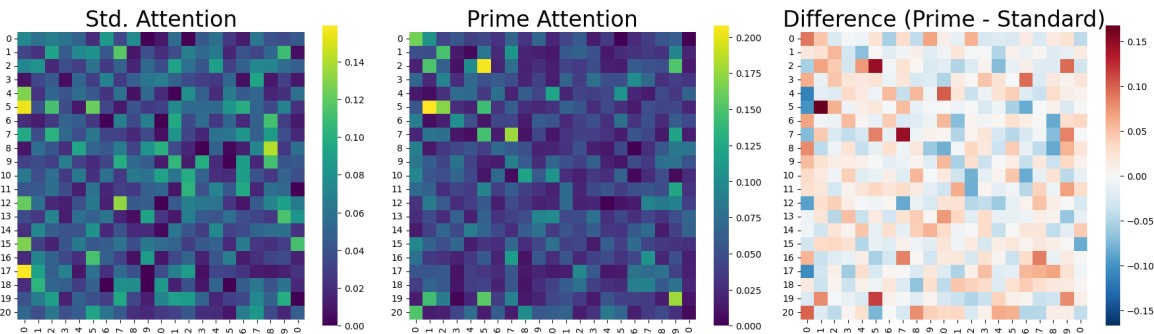

*Figure 7.* Attention maps of standard attention (left), prime attention (middle), and their difference (right) on the Weather dataset. Warmer colors (yellow/green) reflect higher attention coefficient and colder colors (blue/purple) reflect lower attention coefficient.

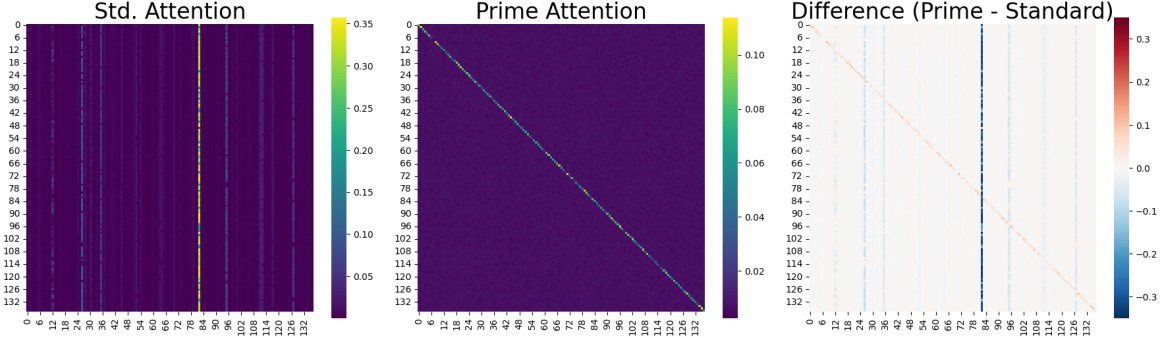

*Figure 8.* Attention maps of standard attention (left), prime attention (middle), and their difference (right) on the Solar dataset. Warmer colors (yellow/green) reflect higher attention coefficient and colder colors (blue/purple) reflect lower attention coefficient.

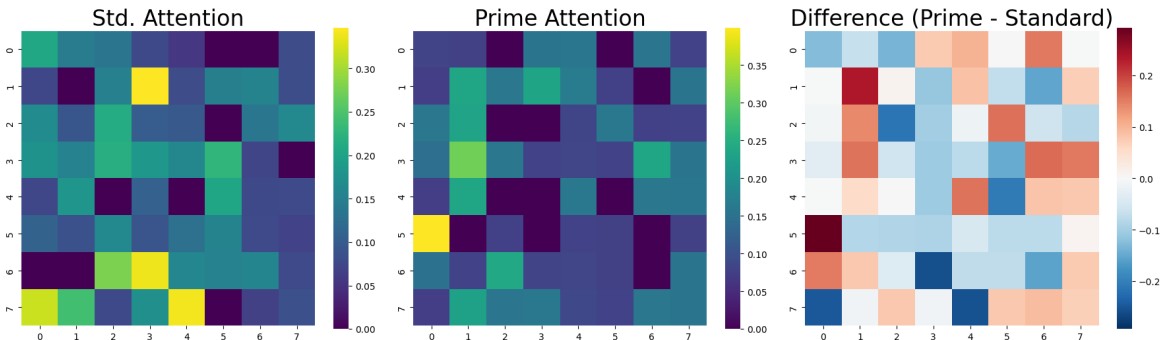

*Figure 9.* Attention maps of standard attention (left), prime attention (middle), and their difference (right) on the Exchange dataset. Warmer colors (yellow/green) reflect higher attention coefficient and colder colors (blue/purple) reflect lower attention coefficient.

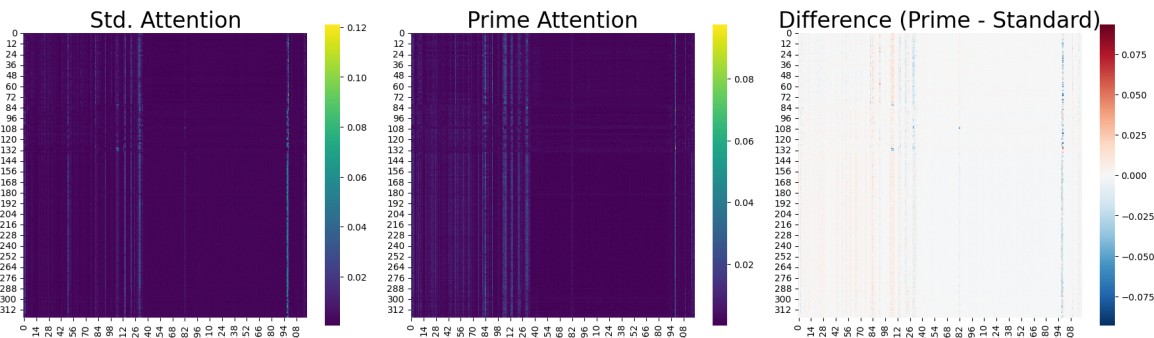

*Figure 10.* Attention maps of standard attention (left), prime attention (middle), and their difference (right) on the ECL dataset. Warmer colors (yellow/green) reflect higher attention coefficient and colder colors (blue/purple) reflect lower attention coefficient.

### E.4. Sequence Length Analysis

Here, we perform a sequence length (or look-back window length $L$) analysis for standard and prime attention. Namely, we are interested in seeing how performance improves with longer sequence length (or more input data). Performance for each model is measured across $L \in \{16, 32, 48, 64, 80, 96\}$ and the prediction horizon is fixed at $H = 96$. We show the results in Figure 1.

Remarkably, prime attention not only shows improved performance at each $L$ as expected, but it also shows that it can match or slightly outperform standard attention's performance with significantly less input sequence length in some instances. To illustrate this, we use a horizontal green dotted line labeled *target threshold* to mark the performance of standard attention when using $L = 96$. As shown in Figures 1a and 1b, prime attention breaks through the target threshold using sequence lengths of 48 and 64, which translates to using 40% and 33% less input data compared to standard attention, respectively. While such significant results were not observed across all datasets, it does align well with the observed benefits of inductive biases in other domains. It also strengthens our argument in Section 5 that prime attention acts as a learnable inductive bias in attention to enable and accelerate effective discoveries of unique relational dynamics across pair-wise interactions. In addition, this finding may be of interest in domains where input data is scarce.

### E.5. KV Filter Modulation Analysis

We include KV analysis to support our decision to use filter modulations on both K and V. As shown in Table 7, while applying filters to only K or only to V do have some advantages, it is substantially more performant to apply filters to both K and V. This supports our intuitive explanations of what prime attention is doing.

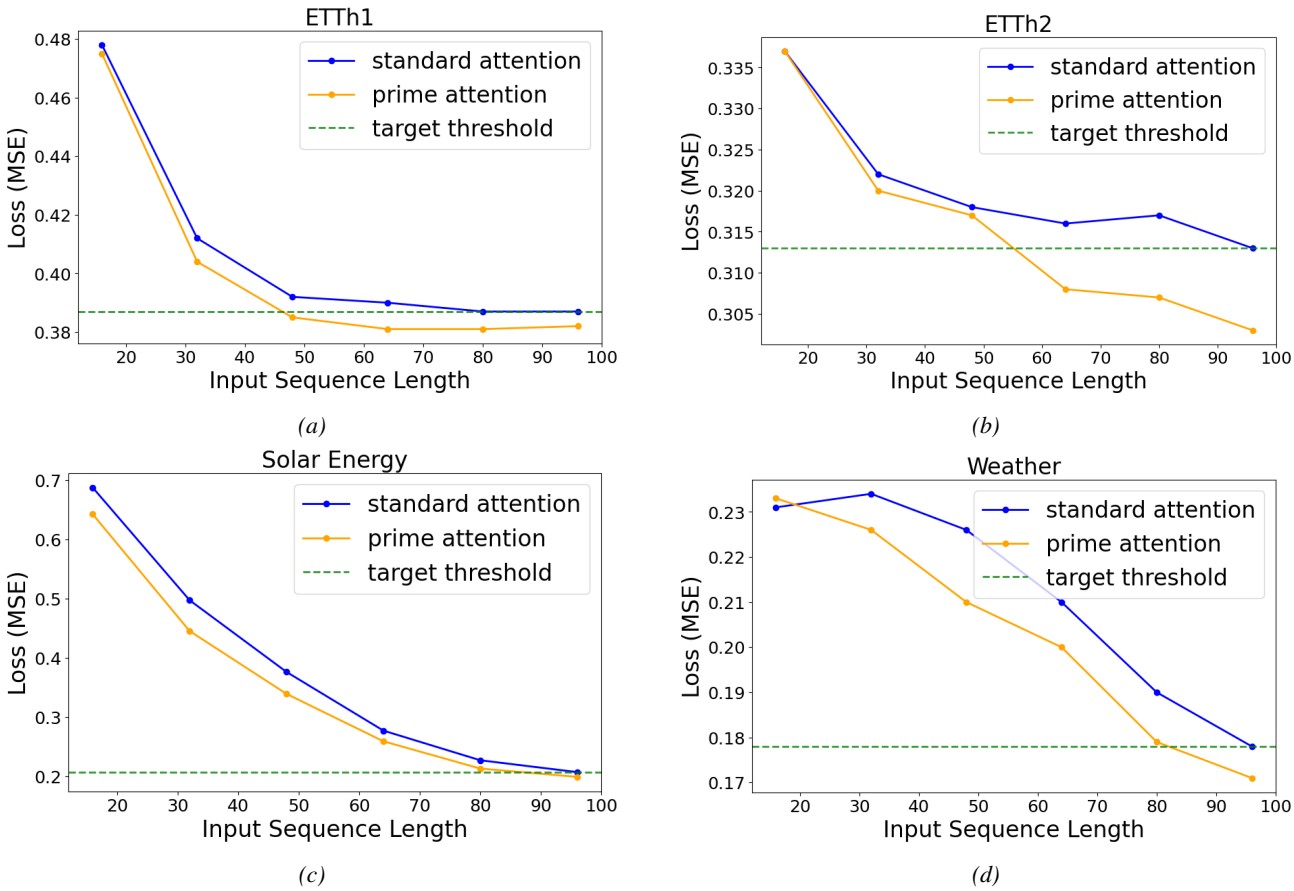

*Figure 11.* Comparing performance between standard (blue) and prime (yellow) attention at various input sequence length (or look-back window $L$). The horizontal target threshold line (green, dotted) represents standard attention's performance at $L = 96$.

*Table 7.* Effect of applying filter modulation to the keys (K), values (V), or both, compared to standard attention. Values are MSE (lower is better); the best result per dataset is in **bold**.

|         | Std. Attn. | K only | V only | K and V |
|---------|------------|--------|--------|---------|
| ETTh1   | 0.387      | 0.384  | 0.385  | **0.379** |
| ETTh2   | 0.315      | 0.304  | 0.310  | **0.296** |
| Weather | 0.178      | 0.174  | 0.174  | **0.170** |

## F. Full Forecasting Results

Unless otherwise stated, forecasting experiments have the look-back window $L$ set to 96 and predictions are made on horizons $H \in \{96, 192, 336, 720\}$ except on the PEMS data, where $H \in \{12, 24, 48, 96\}$. Avg. indicates the averaged loss value of all horizons.

We provide the full comparison between prime attention and existing works in MTS that enhance and leverage relation information in Table 8. The best performance from each row is denoted in **bold**.

We provide the full forecasting results in Table 9. Each model's original baseline performance using standard attention is displayed side-by-side with its updated performance using prime attention. Standard attention refers to the baseline full-attention (or its equivalent) that the transformer model was originally equipped with. To test prime attention's impact on overall transformer model performance, the transformer is first fine-tuned on standard attention and its attention module is subsequently replaced with prime attention only adjusting dropout rate. The better performance (between standard and prime attention) within each model is shown in **teal** when prime attention is better and in **brown** when standard attention is better. Overall, using prime attention outperforms the original baselines consistently across all datasets and models, with up to 6.5% improvement in performance for the Weather dataset.

*Table 8.* Performance comparison between prime attention and existing works in MTS that enhance and leverage relational information. Prime attention is using iTransformer (Liu et al., 2024) as backbone.

| Model | | *Prime Attn.* **(Ours)** | | DIFF Transformer (2025) | | LagTS (2025) | | LIFT (2024) | | Autoformer (2021) | |
| Metrics | | MSE | MAE | MSE | MAE | MSE | MAE | MSE | MAE | MSE | MAE |
|---|---|---|---|---|---|---|---|---|---|---|---|
| ETTh1 | 96 | **0.379** | **0.398** | 0.385 | 0.400 | 0.384 | 0.402 | 0.385 | 0.404 | 0.449 | 0.459 |
| | 192 | 0.432 | **0.428** | 0.438 | 0.431 | 0.437 | 0.435 | **0.431** | 0.429 | 0.500 | 0.482 |
| | 336 | 0.472 | 0.449 | 0.480 | 0.453 | **0.470** | **0.449** | 0.475 | 0.450 | 0.521 | 0.496 |
| | 720 | 0.478 | 0.475 | 0.479 | 0.477 | **0.466** | **0.469** | 0.479 | 0.473 | 0.514 | 0.512 |
| | Avg. | 0.440 | **0.438** | 0.446 | 0.440 | **0.439** | 0.439 | 0.443 | 0.439 | 0.496 | 0.487 |
| ETTh2 | 96 | **0.296** | **0.348** | 0.309 | 0.360 | 0.305 | 0.351 | 0.306 | 0.351 | 0.346 | 0.388 |
| | 192 | **0.374** | 0.399 | 0.385 | 0.405 | 0.381 | **0.399** | 0.390 | 0.406 | 0.456 | 0.452 |
| | 336 | 0.416 | 0.428 | 0.422 | 0.434 | **0.416** | **0.428** | 0.418 | 0.429 | 0.482 | 0.486 |
| | 720 | **0.420** | **0.441** | 0.428 | 0.446 | 0.425 | 0.444 | 0.427 | 0.446 | 0.515 | 0.511 |
| | Avg. | **0.377** | **0.404** | 0.386 | 0.411 | 0.382 | 0.405 | 0.386 | 0.408 | 0.450 | 0.459 |
| Exchange | 96 | 0.083 | 0.201 | **0.083** | **0.201** | 0.085 | 0.206 | 0.085 | 0.205 | 0.197 | 0.323 |
| | 192 | **0.172** | 0.295 | 0.173 | **0.295** | 0.181 | 0.304 | 0.178 | 0.302 | 0.300 | 0.369 |
| | 336 | **0.314** | **0.405** | 0.332 | 0.418 | 0.331 | 0.416 | 0.328 | 0.415 | 0.509 | 0.524 |
| | 720 | 0.786 | 0.668 | 0.839 | 0.690 | **0.768** | **0.663** | 0.831 | 0.691 | 1.447 | 0.941 |
| | Avg. | **0.339** | **0.392** | 0.357 | 0.401 | 0.341 | 0.397 | 0.356 | 0.403 | 0.613 | 0.539 |
| Weather | 96 | **0.170** | **0.210** | 0.176 | 0.215 | 0.173 | 0.216 | 0.172 | 0.216 | 0.266 | 0.336 |
| | 192 | **0.218** | **0.254** | 0.225 | 0.258 | 0.224 | 0.260 | 0.225 | 0.260 | 0.307 | 0.367 |
| | 336 | **0.276** | **0.296** | 0.280 | 0.299 | 0.280 | 0.300 | 0.283 | 0.302 | 0.359 | 0.395 |
| | 720 | **0.351** | **0.347** | 0.356 | 0.349 | 0.355 | 0.349 | 0.358 | 0.350 | 0.419 | 0.428 |
| | Avg. | **0.254** | **0.277** | 0.259 | 0.280 | 0.258 | 0.281 | 0.260 | 0.282 | 0.338 | 0.382 |
| ECL | 96 | 0.148 | 0.239 | **0.147** | **0.239** | - | - | 0.160 | 0.250 | 0.201 | 0.317 |
| | 192 | 0.164 | 0.255 | **0.161** | **0.252** | - | - | 0.183 | 0.267 | 0.222 | 0.334 |
| | 336 | 0.180 | 0.271 | **0.175** | **0.268** | - | - | 0.199 | 0.283 | 0.231 | 0.338 |
| | 720 | **0.208** | **0.297** | 0.210 | 0.299 | - | - | 0.230 | 0.310 | 0.254 | 0.361 |
| | Avg. | 0.175 | 0.266 | **0.173** | **0.265** | - | - | 0.193 | 0.278 | 0.227 | 0.338 |
| Traffic | 96 | **0.392** | **0.264** | 0.395 | 0.265 | - | - | 0.504 | 0.322 | 0.613 | 0.388 |
| | 192 | **0.415** | **0.274** | 0.419 | 0.275 | - | - | 0.499 | 0.319 | 0.616 | 0.382 |
| | 336 | **0.432** | **0.282** | 0.436 | 0.283 | - | - | 0.508 | 0.321 | 0.622 | 0.337 |
| | 720 | 0.465 | 0.300 | **0.464** | **0.299** | - | - | 0.542 | 0.339 | 0.660 | 0.408 |
| | Avg. | **0.426** | **0.280** | 0.429 | 0.281 | - | - | 0.513 | 0.325 | 0.628 | 0.379 |

Table 9. Comparison between standard attention and prime attention on recent state-of-the-art transformer models.

| Model | | Timer-XL (2025) | | | | FreDF (2025) | | | | iTransformer (2024) | | | |
| Attn Type | | Standard | | *Prime* | | Standard | | *Prime* | | Standard | | *Prime* | |
| | | MSE | MAE | MSE | MAE | MSE | MAE | MSE | MAE | MSE | MAE | MSE | MAE |
|---|---|---|---|---|---|---|---|---|---|---|---|---|---|
| ETTh1 | 96 | 0.380 | 0.398 | **0.377** | **0.394** | 0.380 | 0.394 | **0.376** | **0.392** | 0.387 | 0.403 | **0.379** | **0.398** |
| | 192 | 0.432 | 0.429 | **0.428** | **0.424** | 0.434 | 0.426 | **0.430** | **0.423** | 0.439 | 0.432 | **0.432** | **0.428** |
| | 336 | 0.473 | 0.453 | **0.469** | **0.448** | 0.479 | 0.451 | **0.473** | **0.448** | 0.482 | 0.455 | **0.472** | **0.449** |
| | 720 | 0.528 | 0.500 | **0.478** | **0.473** | **0.516** | **0.500** | 0.517 | 0.501 | 0.489 | 0.481 | **0.478** | **0.475** |
| | Avg. | 0.453 | 0.445 | **0.438** | **0.435** | 0.452 | 0.443 | **0.449** | **0.441** | 0.449 | 0.443 | **0.440** | **0.438** |
| ETTh2 | 96 | 0.298 | 0.348 | **0.293** | **0.345** | 0.289 | 0.340 | **0.286** | **0.337** | 0.315 | 0.361 | **0.296** | **0.348** |
| | 192 | 0.383 | 0.402 | **0.376** | **0.398** | 0.382 | 0.400 | **0.380** | **0.395** | 0.380 | 0.402 | **0.374** | **0.399** |
| | 336 | 0.435 | 0.441 | **0.429** | **0.438** | **0.438** | 0.444 | 0.438 | **0.443** | 0.421 | 0.431 | **0.416** | **0.428** |
| | 720 | 0.467 | 0.469 | **0.458** | **0.465** | 0.451 | **0.461** | **0.450** | 0.460 | 0.423 | 0.443 | **0.420** | **0.441** |
| | Avg. | 0.396 | 0.415 | **0.389** | **0.412** | 0.390 | 0.411 | **0.389** | **0.409** | 0.385 | 0.409 | **0.377** | **0.404** |
| ETTm1 | 96 | 0.329 | 0.364 | **0.321** | **0.359** | 0.325 | 0.359 | **0.318** | **0.353** | 0.337 | 0.371 | **0.325** | **0.363** |
| | 192 | 0.394 | 0.403 | **0.384** | **0.397** | 0.374 | 0.384 | **0.368** | **0.382** | 0.377 | 0.392 | **0.368** | **0.388** |
| | 336 | 0.458 | 0.441 | **0.444** | **0.433** | 0.418 | 0.411 | **0.415** | **0.409** | 0.420 | 0.418 | **0.405** | **0.412** |
| | 720 | 0.581 | 0.503 | **0.560** | **0.491** | **0.492** | **0.454** | 0.500 | 0.462 | 0.487 | 0.456 | **0.476** | **0.451** |
| | Avg. | 0.441 | 0.428 | **0.427** | **0.420** | 0.402 | **0.402** | **0.400** | 0.402 | 0.405 | 0.409 | **0.394** | **0.404** |
| ETTm2 | 96 | 0.181 | 0.264 | **0.178** | **0.262** | 0.184 | 0.263 | **0.183** | **0.262** | 0.182 | 0.266 | **0.178** | **0.262** |
| | 192 | 0.248 | 0.307 | **0.245** | **0.306** | 0.250 | 0.304 | **0.249** | **0.303** | 0.247 | 0.308 | **0.243** | **0.304** |
| | 336 | 0.316 | 0.350 | **0.313** | **0.348** | 0.311 | **0.342** | **0.310** | 0.342 | 0.310 | 0.347 | **0.304** | **0.343** |
| | 720 | 0.425 | 0.413 | **0.422** | **0.412** | **0.412** | **0.399** | 0.412 | 0.399 | 0.410 | 0.404 | **0.403** | **0.400** |
| | Avg. | 0.293 | 0.334 | **0.290** | **0.332** | **0.289** | **0.327** | 0.289 | 0.327 | 0.287 | 0.331 | **0.282** | **0.327** |
| Weather | 96 | 0.203 | 0.253 | **0.173** | **0.214** | 0.174 | 0.210 | **0.168** | **0.207** | 0.178 | 0.216 | **0.170** | **0.210** |
| | 192 | 0.225 | 0.292 | **0.219** | **0.257** | 0.227 | 0.257 | **0.220** | **0.253** | 0.227 | 0.259 | **0.218** | **0.254** |
| | 336 | 0.303 | 0.326 | **0.277** | **0.299** | 0.283 | 0.298 | **0.278** | **0.296** | 0.282 | 0.299 | **0.276** | **0.296** |
| | 720 | 0.375 | 0.371 | **0.365** | **0.356** | 0.362 | 0.350 | **0.359** | **0.348** | 0.358 | 0.349 | **0.351** | **0.347** |
| | Avg. | 0.277 | 0.311 | **0.259** | **0.282** | 0.262 | 0.279 | **0.256** | **0.276** | 0.261 | 0.281 | **0.254** | **0.277** |
| Solar-Energy | 96 | 0.200 | 0.235 | **0.197** | **0.234** | 0.210 | 0.226 | **0.203** | **0.220** | 0.207 | 0.235 | **0.199** | **0.226** |
| | 192 | **0.248** | **0.266** | 0.251 | 0.269 | 0.233 | 0.251 | **0.225** | **0.245** | 0.242 | 0.266 | **0.228** | **0.258** |
| | 336 | **0.300** | **0.298** | 0.301 | 0.299 | 0.244 | 0.267 | **0.241** | **0.264** | 0.252 | 0.278 | **0.240** | **0.270** |
| | 720 | 0.378 | 0.338 | **0.365** | **0.336** | **0.247** | **0.270** | 0.247 | 0.273 | 0.249 | 0.279 | **0.247** | **0.278** |
| | Avg. | 0.282 | **0.284** | **0.279** | 0.285 | 0.234 | 0.254 | **0.229** | **0.251** | 0.238 | 0.265 | **0.229** | **0.258** |
| ECL | 96 | **0.136** | **0.230** | 0.136 | 0.230 | 0.143 | 0.233 | **0.142** | **0.231** | **0.146** | 0.240 | 0.148 | **0.239** |
| | 192 | 0.158 | 0.252 | **0.157** | **0.250** | 0.159 | 0.249 | **0.156** | **0.246** | **0.163** | **0.255** | 0.164 | 0.255 |
| | 336 | 0.180 | 0.277 | **0.177** | **0.272** | 0.172 | 0.264 | **0.170** | **0.263** | **0.178** | **0.271** | 0.180 | 0.271 |
| | 720 | 0.242 | 0.333 | **0.232** | **0.321** | 0.203 | **0.293** | **0.202** | 0.293 | 0.209 | 0.300 | **0.208** | **0.297** |
| | Avg. | 0.179 | 0.273 | **0.176** | **0.268** | 0.169 | 0.260 | **0.168** | **0.258** | **0.174** | 0.267 | 0.175 | **0.266** |
| Traffic | 96 | 0.387 | 0.259 | **0.384** | **0.257** | 0.392 | 0.259 | **0.390** | **0.257** | 0.394 | 0.266 | **0.392** | **0.264** |
| | 192 | 0.418 | 0.276 | **0.415** | **0.275** | 0.419 | 0.271 | **0.415** | **0.268** | 0.420 | 0.276 | **0.415** | **0.274** |
| | 336 | 0.448 | **0.290** | **0.447** | 0.299 | 0.435 | 0.279 | **0.430** | **0.276** | 0.436 | 0.284 | **0.432** | **0.282** |
| | 720 | 0.518 | 0.325 | **0.496** | **0.323** | 0.464 | 0.296 | **0.461** | **0.294** | 0.468 | 0.301 | **0.465** | **0.300** |
| | Avg. | 0.443 | **0.288** | **0.436** | 0.289 | 0.428 | 0.276 | **0.424** | **0.274** | 0.430 | 0.282 | **0.426** | **0.280** |
| PEMS03 | 12 | - | - | - | - | 0.068 | **0.171** | **0.067** | 0.171 | **0.069** | **0.174** | 0.070 | 0.174 |
| | 24 | - | - | - | - | **0.095** | 0.204 | 0.095 | **0.202** | 0.098 | 0.208 | **0.096** | **0.206** |
| | 48 | - | - | - | - | 0.155 | 0.262 | **0.151** | **0.257** | 0.159 | 0.269 | **0.158** | **0.264** |
| | 96 | - | - | - | - | 0.232 | 0. 331 | **0.219** | **0.318** | 0.242 | 0.342 | **0.230** | **0.327** |
| | Avg. | - | - | - | - | 0.138 | 0.242 | **0.133** | **0.237** | 0.142 | 0.248 | **0.139** | **0.243** |
| PEMS08 | 12 | - | - | - | - | 0.078 | 0.177 | **0.076** | **0.173** | 0.080 | 0. 181 | **0.078** | **0.177** |
| | 24 | - | - | - | - | 0.112 | 0.210 | **0.107** | **0.204** | 0.116 | 0.217 | **0.111** | **0.210** |
| | 48 | - | - | - | - | 0.185 | 0.267 | **0.174** | **0.258** | 0.196 | 0.281 | **0.183** | **0.267** |
| | 96 | - | - | - | - | 0.312 | 0.344 | **0.288** | **0.327** | 0.338 | 0.373 | **0.297** | **0.339** |
| | Avg. | - | - | - | - | 0.172 | 0.250 | **0.161** | **0.241** | 0.183 | 0.263 | **0.167** | **0.248** |

# G. Forecasting Accuracy Visualization

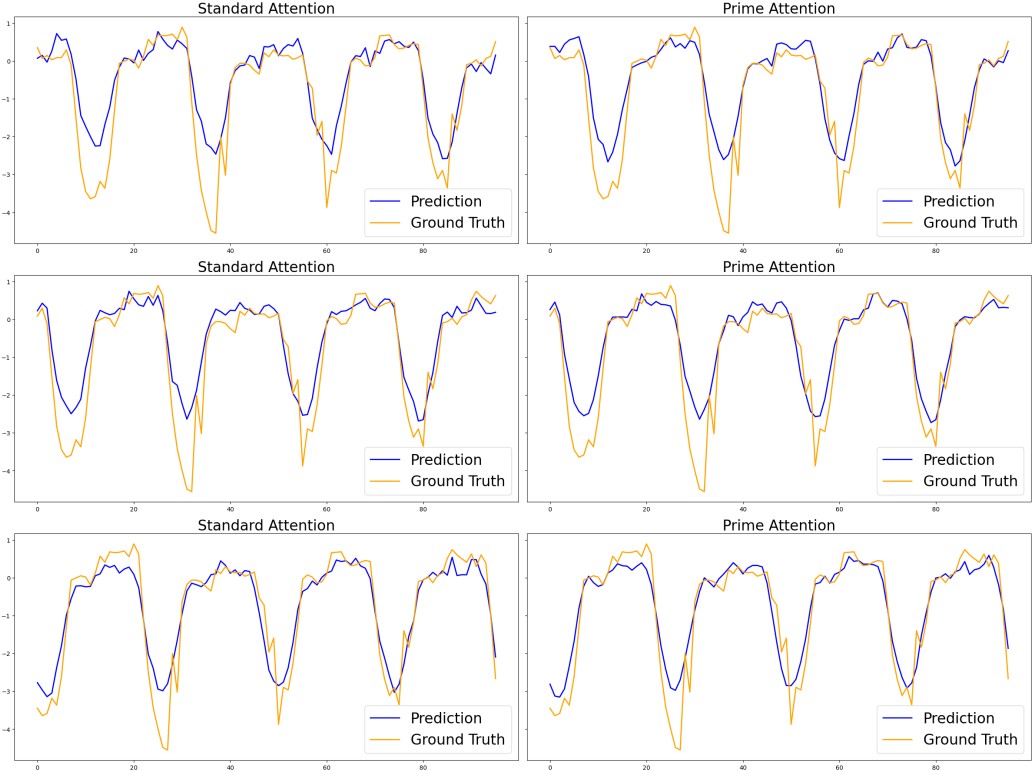

*Figure 12.* Forecasting visualization on ETTh1 dataset using predictions (in blue) made from standard attention (left) and prime attention (right) against ground-truth labels (in yellow). Each row represents a random batch.

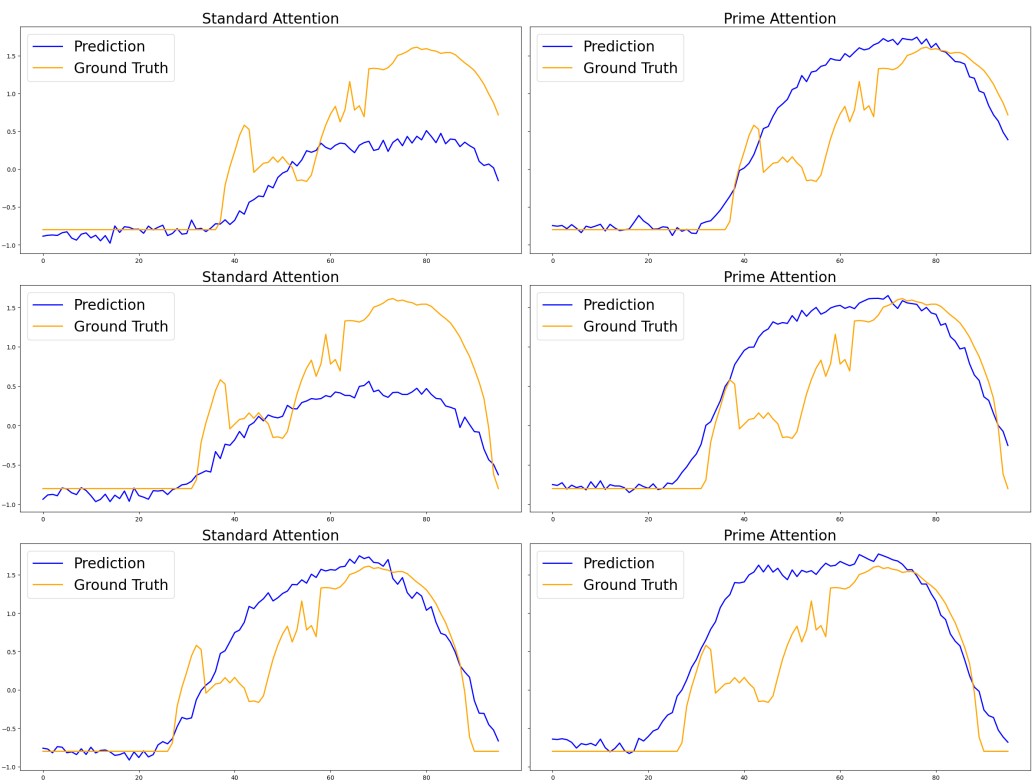

*Figure 13.* Forecasting visualization on Solar dataset using predictions (in blue) made from standard attention (left) and prime attention (right) against ground-truth labels (in yellow). Each row represents a random batch.

