# OpenReview forum: "Dynamic Relational Priming Improves Transformer in Multivariate Time Series"
_ICML.cc/2026/Conference — ICML 2026 regular_

### Official Review · Reviewer_LMTT · 2026-03-12

**Soundness:** 2
**Presentation:** 3
**Significance:** 3
**Originality:** 2
**Overall Recommendation:** 3
**Confidence:** 3

**Summary:**

The authors propose prime attention (attention with dynamic relational priming) in the Transformer to better align the attention mechanism for the requirements of the MTS domain. Dynamic priming modulates token representations for each specific pairwise interaction, optimizing for computational alignment between tokens based on their relationship characteristics. Simulation studies show improvement in forecasting accuracy and less sequence length required compared to standard attention in multivariate time series.

**Compliance With Llm Reviewing Policy:**

Affirmed.

**Final Justification:**

The authors have addressed my concerns, and I will raise the score

**Key Questions For Authors:**

The authors should clarify the difference between other Dynamic Attention in Transformers, like “SDformer: Transformer with Spectral Filter and Dynamic Attention for Multivariate Time Series Long-term Forecasting”.

The authors should also add simulation comparisons with the SDformer.

The improvements in the simulation results are also very limited like Table1,6,7.

**Limitations:**

Did not discuss the limitations.
The improvements are also very limited like Table1,6,7.
The authors should clarify the difference between other Dynamic Attention in Transformers, like “SDformer: Transformer with Spectral Filter and Dynamic Attention for Multivariate Time Series Long-term Forecasting”.

The authors should also add simulation comparisons with the SDformer.

**Strengths And Weaknesses:**

The experiments are not well-designed and they only compare with the standard attention but do not compare with other methods like “SDformer: Transformer with Spectral Filter and Dynamic Attention for Multivariate Time Series Long-term Forecasting”, which also uses Dynamic Attention in a Transformer. The authors should clarify the difference. The improvements are also very limited, as in Tables 1, 6, and 7. The presentation is OK. Overall, the paper addresses a relevant problem for Dynamic Attention in the Transformer.  However, the improvement of the proposed method is very limited.

---

> ### Author Rebuttal · Authors · 2026-03-30
>
> We thank the reviewer for their thorough review. We address reviewer concerns below.
> ### Concern 1: "Experiments only compare with standard attention and are not well designed"
> **This characterization is inaccurate.** We respectfully point out that we do make comparisons against recent SOTA models outside of standard attention in Table 2.
> - **Table 2 (full version: Table 6)**: Directly compares prime attention against **four** independent methods that perform sophisticated relational learning beyond standard attention:
>     - **DIFF Transformer** (Ye et al., ICLR 2025): attention coefficient enhancement via differential attention maps
>     - **LagTS** (Liu et al., ICASSP 2025): explicit lead-lag relationship modeling
>     - **LIFT** (Zhao & Shen, 2024): lead-lag estimation during pre-processing
>     - **Autoformer** (Wu et al., NeurIPS 2021): auto-correlation-based attention replacement
>
> **Regarding SDformer specifically**: We are happy to include SDformer in our comparison table in the revised manuscript. However, we emphasize that their "dynamic attention" is fundamentally different to what we refer to as dynamic learning. **The word "dynamic" in SDformer refers to learnable kernel parameters, not pair-specific representation adaptation like in prime attention**. We will also add SDFormer to related works.
>
> **Regarding "experiments are not well designed"**: We respectfully disagree with this assessment. We would argue that our experimental setups and findings are actually tightly coupled with our hypotheses.
> - sequence length experiment (fig 1): is **directly related to our inductive bias hypothesis from section 5 and shows to what degree prime attention's pair-wise inductive bias can outperform standard attention** even when using much less input data.
> - attention map analysis (fig 2,3): shows how prime attention learns fundamentally different patterns than standard attention. **This complements our numerical results from Table 1 and shows that prime attention isn't just achieving better performance, but rather learns fundamentally distinct patterns that are more reliable**.
> - comparing against standard attention with additional attention head (Table 3): by comparing prime attention against standard attention with an additional attention head and around 35% more parameters, we confirm that just introducing more computation and parameters do not lead to improved performance. **This shows that prime attention is addressing a fundamental performance bottleneck of standard attention and implies that prime attention learns fundamentally more informative patterns than standard attention**, especially in heterogeneous datasets.
> ### Concern 2: "clarify difference from SDFormer's attention"
> SDformer's DDA operates on attention _scores_; Prime attention operates on token _representations_.
>
> SDformer's Dynamic-Directional-Attention (DDA) applies a kernel function to Q and K vectors before computing the attention score. The goal is to make the resulting attention score distribution less uniform (i.e., higher Gini coefficient). Critically:
> - The Q and K vectors are still **static**: the same Q_i is used against all K_j, and the same K_j is used against all Q_i. The kernel function is applied independently to each vector without any pair-specific conditioning.
> - DDA modulates the **attention coefficients** not the underlying token representations.
> - Under our formal framework (Definition 3.1), SDformer performs **static relational learning**
>
> **Proposed revision**: We will add an explicit discussion of SDformer in the related work section of the revised manuscript to make this distinction clearer.
> ### Concern 3: "Improvements are very limited"
> We respectfully disagree with this characterization for several reasons:
> - The improvements are consistent and significant for the MTS domain
> 	- **Up to 6.5% improvement** on Weather with Timer-XL: this is a substantial gain in MTS forecasting where 1-2% improvements are generally seen as statistically significant.
> 	- **Average 4.0% improvement** on heterogeneous datasets (Solar, Weather) across all backbone models.
> 	- **Average 5.5% improvement** on short-term PEMS forecasts.
> - The improvements come at negligible cost: prime attention adds only ~1.5% additional parameters on top of standard attention. Methods that achieve larger gains often introduce substantially more parameters or architectural complexity.
> - Context within the field:
> 	- LagTS (ICASSP 2025) reports comparable gains
> 	- FreDF (ICLR 2025) reports comparable gains
> 	- SDFormer (IJCAI 2024) which the reviewer cites, reports comparable gains
> ### Concern 4: "Did not discuss limitations"
> We acknowledge this and will add an explicit limitations section. However, we note that several limitations are already discussed throughout the paper:
> 1. Complexity overhead: Section 4 and section E.2
> 2. Performance on homogeneous datasets: section 6.1
> 3. Scalability: section 4

---

> > ### Author Rebuttal · Reviewer_LMTT · 2026-04-02
> >
> > The authors have addressed my concerns, and I will raise the score

---

> > > ### Author Response · Authors · 2026-04-02
> > >
> > > We sincerely thank the reviewer for their constructive engagement and for confirming that all concerns have been fully resolved. We are glad that the clarification of the fundamental distinction between prime attention's dynamic relational learning and attention-coefficient-sharpening methods (including SDformer), the broader experimental comparisons in Table 2/6, and the discussion of limitations addressed the reviewer's questions. We will incorporate all suggested revisions (including an explicit SDformer comparison and a dedicated limitations section) into the updated manuscript, which we believe has meaningfully strengthened the paper. We thank the reviewer again for their time and feedback.

---

### Official Review · Reviewer_fdvM · 2026-03-12

**Soundness:** 2
**Presentation:** 3
**Significance:** 3
**Originality:** 3
**Overall Recommendation:** 5
**Confidence:** 4

**Summary:**

This paper propose a new attention mechanism to make the key and query vector representation different between each pair. The question the paper investigate is: Can pair-specific modulation of token representations inside attention better model heterogeneous inter-channel relations in MTS than standard attention. Their core claim is that standard attention does static relational learning. For a token, its representation does not change depending on which other token it interacts with in that layer. The paper argues this is a bad fit for MTS, where one channel-pair may exhibit lead-lag structure and another may have instantaneous correlation. Their experiments result show better forecasting error and some sequence-ength efficiency gains.

**Compliance With Llm Reviewing Policy:**

Affirmed.

**Final Justification:**

The rebuttal addressed most of my concerns about statistical robustness and failure case analysis. For the concern that Table 2 is incomplete, the authors’ explanation makes sense to me. The only issue is that they did not provide the corresponding experiments in the rebuttal. Personally, I think this is acceptable given the short rebuttal period. However, the table should definitely be expanded in the final version of the paper, as promised by the authors. All other responses resolved my concerns, so I raised my score to 5.

**Key Questions For Authors:**

* I may have overlooked something, but is there any reason why the results on the PEMS dataset are missing for Timer-XL?

* Can the authors also discuss how the proposed method relates to existing gated attention work such as [3]?

[3] Qiu, Z., Wang, Z., Zheng, B., Huang, Z., Wen, K., Yang, S., ... & Lin, J. (2025). Gated attention for large language models: Non-linearity, sparsity, and attention-sink-free.

**Limitations:**

The paper acknowledges some limitations, but the discussion could be more explicit. A brief discussion of failure cases or settings with smaller gains would improve the paper.

**Strengths And Weaknesses:**

### Strength

* The paper is well-written and easy to understand.
* The proposed method is simple, clean, supported with formal formulation and shows improvement on the wide-range od dataset.
* The paper further conduct analysis on the method's efficiency (better data efficiency on some dataset) and the attention map.



### Weakness

* The paper reports results averaged over 5 seeds, but it does not report standard deviations. Including them would help assess robustness of the method.

* The paper does not discuss the failure cases in the detailed results of Table 6 in the appendix. Beyond showing where the method works, it is also important to analyze where it performs poorly. For example, what time-series patterns or data characteristics make the method less effective? Why LagTS perform better in some cases?

* The comparision to existing works in MTS that enhance and leverage relational information is only tested on 4 datasets. Table 2 to me need to be test on more dataset. Is there any reason why the author didn't test on the same datasets include in the table 2?

#### Minor suggestions
* The discussion of universal approximation could cite more related theoretical work on transformers and attention, for example:
[1] Kajitsuka, T., & Sato, I. (2023). Are transformers with one layer self-attention using low-rank weight matrices universal approximators?.
[2] Hu, J. Y. C., Liu, H., Chen, H. Y., Wu, W., & Liu, H. (2025). Universal approximation with softmax attention.

---

> ### Author Rebuttal · Authors · 2026-03-30
>
> We thank the reviewer for their thorough review. We address reviewer concerns and introduce some new results below.
> ### W1
> We provide a snippet of the full results (in MSE) below (using iTransformer as backbone and using random seed in each run for 5 runs).
>
> | Horizon \ Dataset | ETTh1            | ETTh2            | Weather          | ECL              | Traffic          |
> | ----------------- | ---------------- | ---------------- | ---------------- | ---------------- | ---------------- |
> | 96                | $0.378\pm 0.001$ | $0.297\pm0.001$  | $0.170\pm0.001$  | $0.146\pm0.0005$ | $0.395\pm0.0001$ |
> | 192               | $0.432\pm0.001$  | $0.376\pm0.001$  | $0.218\pm0.001$  | $0.164\pm0.001$  | $0.417\pm0.0002$ |
> | 336               | $0.473\pm0.001$  | $0.413\pm0.001$  | $0.275\pm0.001$  | $0.179\pm0.001$  | $0.431\pm0.0001$ |
> | 720               | $0.479\pm0.001$  | $0.420\pm0.0008$ | $0.354\pm0.0005$ | $0.214\pm0.002$  | $0.463\pm0.001$  |
> | Avg.              | $0.441\pm0.001$  | $0.377\pm0.001$  | $0.254\pm0.0008$ | $0.176\pm0.001$  | $0.427\pm0.0003$ |
> ### W2
> We appreciate this feedback and provide expanded failure case analysis. Prime attention shows the smallest gains on ECL and Traffic, where all channels record the same physical attribute (electricity consumption and road occupancy, respectively). This homogeneity means standard attention's static relational learning is already well-matched to the data (per our hypothesis). Conversely, on Weather and Solar where channels represent fundamentally different physical quantities, gains are substantial (up to 6.5%). Overall, when standard attention is already compatible with the given data, prime attention struggles beyond marginal improvements and can be overkill.
>
> Regarding LagTS outperforming at longer horizons on ETTh1: LagTS employs a specialized loss function explicitly designed to guide lead-lag learning, giving it an advantage when lead-lag is the dominant relational pattern in a low-dimensional system (7 channels). Prime attention's advantage is generality. It discovers diverse relational patterns without hardcoding any specific type, which becomes increasingly beneficial as channel heterogeneity grows.
> ### W3
> We agree that Table 2 should be expanded. Currently, Table 2 only uses datasets that were reported by LagTS. Because LagTS didn't offer any source code, we could not faithfully duplicate their results on other datasets that they did not use.
>
> **Proposed revision**: We can keep LagTS results fixed and expand Table 2 only for other models like our implementation of the differential transforemr and LIFT. Please let us know if this is a direction you would like us to go in.
> ### Questions
> - We had trouble running Timer-XL on the PEMS datasets from their official codebase. They did not use PEMS data in their paper and we had some trouble getting PEMS to show reasonable performance on their official models so we decided to leave them blank.
> - Gated attention (Qiu et al., 2025) introduces non-linear gating mechanisms that modulate attention scores or outputs, effectively controlling how much information flows through each attention pathway. While this improves upon standard softmax attention, it still operates on static token representations. The key and value vectors presented by each token remain invariant across all interaction partners. In our framework (Section 3), gated attention falls under static relational learning with enhanced coefficients, similar to differential attention and selective attention discussed in Section 2. Prime attention operates at a fundamentally different level: it modulates the token representations themselves on a per-pair basis before scoring occurs. This distinction is crucial for MTS where the type of information relevant between channels varies (not just the amount), requiring different representational views rather than different attention weights on the same view. **We will include this to our related works section.**

---

> > ### Author Rebuttal · Reviewer_fdvM · 2026-04-02
> >
> > Thanks for the detailed response. My concern on W1, W2 has been resolved. I suggest the author include those changes into final revision.
> >
> > For W3, thanks for the clarification, I think the proposed direction looks good to me.
> >
> > For Q1, can the author comment more on what does "reasonable performance" mean? What is the exact issue here?

---

> > > ### Author Response · Authors · 2026-04-02
> > >
> > > Thank you for your response. We will include all suggested changes to our final revision.
> > >
> > > For Q1, we apologize for the lack of clarity. We experienced some weird issues with Timer-XL on PEMS data such as getting nans on train/val losses as well as the final results being orders of magnitude higher in error than those of other models (we checked for normalization and that was not the issue). For example, for PEMS03, the expected test loss is around 0.07 MSE but we would see numbers greater than 1.0. For PEMS07, we got OOM with Timer-XL. These issues were occuring with their base model before introducing prime attention so we decided to steer clear of it.
> > >
> > > In addition, we wanted to let you know that we will strengthen the citations around universal approximation using reviewer suggested papers. Thank you for this.
> > >
> > > Overall, we wanted to once again thank the reviewer for a constructive and positive review. Your review has allowed us to substantially clarify and strengthen the paper.

---

### Official Review · Reviewer_JjLC · 2026-03-13

**Soundness:** 3
**Presentation:** 4
**Significance:** 4
**Originality:** 3
**Overall Recommendation:** 3
**Confidence:** 3

**Summary:**

This paper introduces Dynamic Relational Priming, implemented through a modified attention mechanism called prime attention. The key idea is to learn pair-specific modulation vectors that adjust key and value representations for each token pair during attention computation. The authors evaluate prime attention on several multivariate time series forecasting benchmarks and show that replacing standard attention with prime attention can improve prediction performance across multiple transformer-based models.

**Compliance With Llm Reviewing Policy:**

Affirmed.

**Ethical Review Concerns:**

N/A.

**Final Justification:**

I still decided to keep my score. Please see my comments.

**Key Questions For Authors:**

See weakness.

**Limitations:**

See weakness.

**Strengths And Weaknesses:**

## Strengths:
1. Modeling heterogeneous dependencies between channels is a well-known difficulty in MTS forecasting.
2. The proposed mechanism offers an intuitive interpretation: enabling token representations to adapt depending on the interaction partner.
3. The experimental results show performance improvements across multiple datasets.

## Weakness:
1. The most significant practical limitation is the memory complexity of $\mathcal{F}$, and this problem will be harder under multi-head settings (increased by the head number). Although the paper claims that the parameter overhead is relatively small, the runtime and memory implications of computing pairwise primers are not thoroughly analyzed. How does the approach scale with sequence length? Does it introduce quadratic overhead beyond standard attention? and how it performs in very long time series settings?
2. The $\mathcal{F}$ can be seen as a "relational memory" or a "channel-conditioned features" during attention computation. If it is only applied to Key (K), this formulation would be transformed into the standard paradigm of Q-former. Did you apply it only to K or only to V?
3. The paper proposes an FFT-based lead–lag initialization strategy (Eq. 13) to bias the relational primers using estimated temporal correlations. However, it is not entirely clear whether this initialization was used in the main experimental results or only presented as an optional component.
4. While the paper shows consistent improvements across several datasets, the analysis does not clearly explain when or why prime attention provides the largest gains. Are improvements primarily observed in datasets with strong heterogeneous channel relationships? Does the method help more for long-range dependencies or short-term correlations?

---

> ### Author Rebuttal · Authors · 2026-03-30
>
> We thank the reviewer for their constructive review. We address reviewer feedback and concerns below.
> ### W1
> We would like to clarify a key architectural detail: the primers F are computed **once per transformer block** and **shared across all attention heads** within that block (Section 4, Implementation Details). They are not replicated per head, so the multi-head setting does not multiply the memory cost of F by the number of heads.
>
> Critically, in the channel-wise attention paradigm (iTransformer, Timer-XL, FreDF), N represents the **number of channels**, not the sequence length. F ∈ R^{N²×d_model} therefore scales quadratically in channels, not in time steps. We demonstrate practicality even on Traffic (N=862) and PEMS (N=358) on a single A100 GPU.
>
> Regarding scalability to very high N: our sparsity ablation (Section E.1, Figure 4) shows that even assigning primers to just 20% of token pairs can outperform standard attention, providing a direct mechanism for controlling memory in high-dimensional settings. Furthermore, Section E.2 (Figure 6) demonstrates that prime attention at 60% adjacency density matches standard attention at full density while using **less memory and less computation time** showing that the effective complexity of prime attention can be lower than standard attention.
>
> We provide **new resutlts** of prime attention's performance (in MSE) vs. memory usage on the Weather dataset below at various channel connectivity densities.
>
> |Adj. Density|Memory|Performance|
> |---|---|---|
> |0.2|765.21 MB|0.180|
> |0.4|1212.75 MB|0.179|
> |0.6|1691.39 MB|0.177|
> |0.8|2177.59 MB|0.176|
> |1.0 (full adj)|2664.64 MB|0.175|
>
> For reference, standard fully-connected attention achieves 0.177 MSE using 2102.92 MB in this setup, **requiring around 400 MB MORE than prime attention** for comparable performance.
>
> **Proposed revision**: we will add an explicit table in the revision showing per-dataset memory overhead of F, clarifying the shared-across-heads design, and discussing the sparsity mechanism as a practical scaling strategy
> ### W2
> Thank you for this insightful observation. We appreciate the connection to Q-former architectures. However, there is a fundamental distinction: Q-former uses cross-attention where queries attend to a separate external source, whereas prime attention is a _self-attention_ mechanism where pair-specific primers modulate representations within the same token set. The modulation in prime attention is _relational_ (specific to each token-pair), not source-specific.
>
> We apply $\mathcal{F}$ to **BOTH** K and V. This design ensures that pair-specific information influences both the attention scoring (through K) and the value aggregation (through V). If $\mathcal{F}$ only modulated K, the model could learn where to attend in a pair-aware manner, but the information extracted (V) would still be static.
> ### W3
> We appreciate this question and apologize for the lack of clarity. **All main experimental results** (Tables 1, 2, 6, 7) use the **full initialization** strategy combining lead-lag estimation and instantaneous correlation. We will make our experimental setups more explicit in our revised paper.
> ### W4
> Analyzing our results systematically reveals three clear patterns:
>
> 1. Channel heterogeneity is the primary driver of improvement. Datasets where channels record qualitatively _different_ physical quantities show the largest gains. This directly validates our hypothesis that heterogeneous inter-channel dynamics benefit from pair-specific representation modulation.
> 2. Short-term forecasting benefits more than long-term. Prime attention achieves 5.5% average improvement on PEMS short-term benchmarks (H ∈ {12, 24, 48, 96} at 5-minute intervals), compared to smaller but consistent gains on long-horizon forecasts. This suggests prime attention excels at capturing fine-grained, noisy relational patterns that are critical for short-term accuracy but may be averaged out over longer horizons.
> 3. Data-scarce regimes amplify the advantage. Section 6.2 shows prime attention's performance gap over standard attention is largest at shorter input sequences, where standard attention lacks sufficient data to learn effective relational patterns. This aligns with our inductive bias argument (Section 5) and has practical implications for domains with limited data.
>
> We will add a dedicated "When Does Prime Attention Help?" subsection in the revision summarizing these findings as practical deployment guidance.

---

> > ### Author Rebuttal · Reviewer_JjLC · 2026-04-02
> >
> > Thank you for the rebuttal. Some of my concerns have been addressed, but the rebuttal still lacks adequate experimental support, especially for W2 and W4. The absence of these experiments makes the response less persuasive and limits its ability to further substantiate the authors’ claims and strengthen the overall contribution of the work.
> >
> > Therefore, I maintain my original score.

---

> > > ### Author Response · Authors · 2026-04-02
> > >
> > > We thank the reviewer for their constructive and thorough feedback. We introduce additional experiments to target remaining concerns.
> > > ### W2
> > > We include key ablations to support our decision to use filter modulations on both K and V. As empirical evidence shows, while applying filters to only K or only to V do have some advantages, it is substantially more performant to apply filters to both K and V. This supports our intuitive explanations of what prime attention is doing.
> > >
> > > |         | std. attn. | K only | V only | K and V |
> > > | ------- | ---------- | ------ | ------ | ------- |
> > > | ETTh1   | 0.387      | 0.384  | 0.385  | 0.379   |
> > > | ETTh2   | 0.315      | 0.304  | 0.310  | 0.296   |
> > > | Weather | 0.178      | 0.174  | 0.174  | 0.170   |
> > > ### W4
> > > We take the weather dataset to isolate the effects of prime attention as it pertains to heterogeneity. For background, the weather dataset contains many pressure-related channels as well as temperature-related channels. For **prime (homogenous)**, **we enable prime attention only within homogenous channels (for example between pressure sensors)**. For **prime (heterogenous)**, **we enable prime attention only for heterogenous channel pairs (such as between pressure and temperature sensors)**. We provide standard attention and the regular full prime attention for reference. We see clearly that while prime attention on homogenous channel pairs also improve performance compared to standard attention, the effect is far greater when prime attention is applied to heterogenous channel pairs. In fact, applying prime attention only for heterogenous channel pairs resulted in comparable performance compared to prime (full), implying that heterogenous channel mixing are where prime attention does the most heavy lifting.
> > >
> > > |         | std. attn | prime (homogenous) | prime (heterogenous) | prime (full) |
> > > | ------- | --------- | ------------------ | -------------------- | ------------ |
> > > | Weather | 0.178     | 0.173              | 0.171                | 0.170        |
> > > As for long term vs. short-term dependencies, we believe that prime attention tends to perform better on short-term data. This is seen in our PEMS experiments where we steadily maintain an average of 5% improvement when using prime attention.
> > >
> > > Overall, we hope these new experiments fully address remaining reviewer concerns. We will aim to expand these experiments (i.e. applying on more datasets) for the revised manuscript and include them in our results section or the appendix.

---

### Official Review · Reviewer_SV58 · 2026-03-13

**Soundness:** 3
**Presentation:** 3
**Significance:** 2
**Originality:** 2
**Overall Recommendation:** 4
**Confidence:** 3

**Summary:**

This paper proposes Prime Attention, a modified transformer attention mechanism designed to better capture heterogeneous inter-channel relationships in multivariate time series (MTS). The key idea is to introduce a pair-specific modulation mechanism, called primers, which dynamically adjusts how one token is represented when interacting with another token.

**Compliance With Llm Reviewing Policy:**

Affirmed.

**Final Justification:**

I think this work empirically and theoretically provide some interesting conclusions. I choose to maintain my score of 4 (I support this work but would be quite ok if this work is rejected)

**Key Questions For Authors:**

The same as the weakness

**Limitations:**

The authors should discuss more on the social impact of the deployment of the time series models, such as model misuse, over-reliance, and failure risks in real-world forecasting pipelines.

**Strengths And Weaknesses:**

Strength:
1. Clear motivation and intuitive method. The proposed method (pair-specific modulation of keys and values through a learnable primer) is straightforward, easy to understand, and architecturally compatible with standard transformer blocks.
2. Broad empirical evaluation across datasets and backbones. The paper evaluates the approach on enormous datasets and considers multiple recent MTS transformer backbones, including Timer-XL, FreDF, and iTransformer.
3. Additional analyses beyond a single benchmark table. I appreciated that the paper does not stop at headline forecasting metrics. It also includes sequence-length efficiency analysis, attention-map visualizations, and a parameter-effectiveness comparison against simply increasing the number of heads.

Weakness:
1. The theoretical framing feels more definitional than genuinely explanatory. A major portion of the paper is built around the distinction between static and dynamic relational learning. However, these notions are introduced by the authors themselves and then used to prove that standard attention is “static” and prime attention is “dynamic.” While internally consistent, these propositions do not yet establish that the proposed distinction is the right one for explaining performance differences in practice.
2. The causal interpretation of the attention-map analysis is too strong. The discussion around standard attention “prematurely converging to an easy solution” or creating a “single point of failure” from attention visualizations is interesting but speculative. Attention maps alone are weak evidence for such claims, especially without perturbation tests, causal ablations, or failure-mode analyses showing that dependence on a highlighted channel truly drives brittleness.
3. The methodological novelty is more incremental than the paper suggests. he proposed operation is essentially a pair-specific multiplicative modulation of keys/values using a learnable primer $F_{ij}$. While the paper frames this as a new category of “dynamic relational learning,” the actual mechanism is still a fairly direct modification of attention. This paper may need to discuss more on the difference from the existing conditional attention, and relation-aware attention ideas.

---

> ### Author Rebuttal · Authors · 2026-03-30
>
> We thank the reviewer for their thorough and positive review. We address their feedback below.
> ### W1
> We totally understand this point and aim to strengthen our theoretical framing.
>
> **a. The distinction is empirically grounded, not merely definitional.**
>
> The static/dynamic distinction makes a falsifiable prediction: if heterogeneous inter-channel relationships are indeed poorly served by static representations, then prime attention should show its largest gains on datasets with heterogeneous channels and minimal gains on homogeneous ones. This is exactly what we observe:
> - Heterogeneous datasets: prime attention achieves **up to 6.5% improvement** (Table 1).
> - Homogeneous datasets: prime attention shows only **marginal improvements** (Table 1).
>
> **b. The formalization enables principled method design.**
>
> Definitions 3.1 and 3.2 are not arbitrary. They isolate the specific computational property (pair-invariance vs. pair-specificity of token representations) that determines whether an attention mechanism can, in principle, adapt to heterogeneous relationships.
>
> **Proposed revision:** We will add explicit discussion connecting the static/dynamic distinction to the observed heterogeneity-dependent performance pattern, making the explanatory (not merely definitional) nature of the framework more prominent.
> ### W2
> We fully acknowledge some interpretations and our wording around attention maps are too strong and we will tone them down in our revision. However, we want to highlight that our claims are supported by more evidence than attention maps alone:
>
> The Solar dataset attention maps (Figure 2) are not presented in isolation. They are accompanied by a 3.8% MSE improvement from prime attention. The fact that prime attention both (a) eliminates the single-channel over-reliance pattern and (b) achieves meaningfully better forecasting accuracy provides some degree of convergent evidence, though we still agree with the reviewer that our wordings are too strong.
>
> **Proposed revision:**
> - Soften the language around our attention map analysis, and remove any over-claiming.
> - Add a note acknowledging that attention map analysis provides correlational rather than causal evidence.
> ### W3
> We respectfully disagree that the contribution is incremental, and we clarify both the novelty and the distinction from prior work:
>
> **a. Simplicity ≠ incrementality.**
>
> The value of prime attention lies not in architectural complexity but in the principle it embodies: that pair-wise relational modeling in attention should condition token representations on the specific interaction partner. This principle, despite its simplicity in implementation, is novel in the MTS attention literature and yields consistent, meaningful improvements.
>
> In addition, a modification that reliably improves Timer-XL, FreDF, and iTransformer across 11 diverse datasets with only ~1.5% additional parameters demonstrates broad utility. Few proposed attention modifications achieve this level of consistent, backbone-agnostic improvement.
>
> **b. Prime attention is fundamentally different from conditional attention and relation-aware attention.**
>
> | Aspect                          | Conditional Attention                                | Relation-Aware Attention (e.g., edge features in GNNs) | Prime Attention                                                               |
> | ------------------------------- | ---------------------------------------------------- | ------------------------------------------------------ | ----------------------------------------------------------------------------- |
> | **What is modulated**           | Attention _coefficients_ (the α values)              | Attention _scores_ or message aggregation              | Token _representations_ (keys and values) before scoring                      |
> | **Granularity**                 | Global or context-level conditioning/gating          | Often type-level (e.g., edge type in R-GCN)            | Pair-specific, continuous modulation                                          |
> | **Effect on representations**   | Tokens remain static; only their attn weights change | Tokens remain static; relational info is additive      | Tokens become dynamic (different representation for each interaction partner) |
> | **Dynamic relational learning** | No (static representations)                          | No (representations remain static)                     | Yes                                                                           |
> We will add more detailed discussions in the related works section of our revised paper.

---

> > ### Author Rebuttal · Reviewer_SV58 · 2026-04-02
> >
> > Thanks for the detailed response. I will choose to maintain my score.

---

> > > ### Author Response · Authors · 2026-04-02
> > >
> > > We thank the reviewer for confirming that our response fully addresses reviewer concerns. We will incorporate all suggested revisions into the final manuscript. We are grateful for the constructive feedback that has improved our work. Thank you.

---

### Decision · Program_Chairs · 2026-04-30

**Decision:**

Accept (regular)

**Comment:**

This paper introduces a novel transformer attention mechanism designed to better capture heterogeneous inter-channel relationships in multivariate time series forecasting. Reviewers acknowledged the paper’s motivation and experimental validation.
However, they also raised concerns regarding the formal definition of dynamic relational learning, insufficient results and analysis, lack of failure cases, among other issues. In addition, the discussion or comparison with previous study on channel dependence modeling in long term  time series forecasting.
During the rebuttal phase, the authors provided additional results and explanations, and committed to improving the paper in line with the reviewers’ suggestions. Most reviewers were satisfied with the authors’ response.
Given the substantial revision workload required, the area chair recommends a weak acceptance.